# Mechanisms of surface solar irradiance variability under broken clouds

Wouter Mol[1] and Chiel van Heerwaarden[1]

[1]Meteorology & Air Quality Group, Wageningen University & Research

**Correspondence:** Wouter Mol (wbmol@wur.nl)

**Abstract.** Surface solar irradiance variability is present under all broken clouds, but the patterns, magnitude of variability, and driving mechanisms vary greatly with cloud type. In this study, we performed numerical experiments to understand which main mechanisms drive surface solar irradiance (SSI) variations across a diverse set of observation-based cloud conditions. The results show that four mechanisms capture the essence. We find that for optically thin ($\tau < 6$) clouds, scattering in the forward
direction (*forward escape*) dominates. In cloud fields with enough optically thin area, such as altocumulus, *forward escape* alone can drive areas of irradiance enhancement of over 50 % of clear-sky irradiance. For flat, optically thick clouds ($\tau > 6$), irradiance is instead scattered diffusely downward (*downward escape*), and (extreme) enhancements are thus found directly below the cloud. For vertically structured clouds, *side escape* dominates domain-averaged diffuse irradiance enhance until the sides become shaded by anvil clouds. Lastly, under optically thick cloud cover, surface albedo enhances radiative fluxes due
to multiple scattering between surface and cloud. This brightens shadows and contributes 10 to 60 % of the total irradiance enhancement for low (0.2) to high (0.8) albedo. With these four mechanisms, we provide a framework for understanding the vast diversity and complexity found in surface solar irradiance and cloudiness. A next step is to apply this analysis to multi-layered cloud fields and non-isolated deep convective clouds.

## 1 Introduction

In this study, we aim to determine the main mechanisms that drive the spatial and temporal patterns of surface solar irradiance in the presence of clouds. Understanding how different clouds and cloud fields create irradiance variability will help simplify the vast diversity and complexity found in observations of surface irradiance and cloudiness.

The search for this simplified understanding is motivated by the numerous impacts of (extreme) variability in surface solar irradiance on the spatiotemporal scales associated with clouds and the inability to resolve this variability in atmospheric models
despite their increasingly high resolution. Many of the impacts relate to non-linear processes that are part of a complex system of feedbacks and interactions, and therefore it matters how solar irradiance is distributed over time, space, and wavelength.

For example, boundary-layer cumulus clouds form as a result of a land surface heated through solar irradiance, but they in turn change the amount and distribution of irradiance and thereby influence their own evolution (e.g., Gristey et al., 2020; Tijhuis et al., 2024). Such a land surface may be full of vegetation doing photosynthesis in response to solar irradiance, taking
up carbon and releasing moisture through stomata, small openings on the surface of leaves. Stomata, however, only slowly open

and close in response to irradiance, thus photosynthesis depends on how much irradiance varies (Way and Pearcy, 2012). The quality of solar irradiance, namely the amount of diffuse (i.e., scattered) radiation and spectral composition, is also influenced by clouds. Ecosystem photosynthesis increases when irradiance is relatively diffuse (Roderick et al., 2001; Gu et al., 2003), impacting land-atmosphere interactions (Vilà-Guerau de Arellano et al., 2023), with the impact of changes in the spectral composition of radiation still unclear (Durand et al., 2021; Huber et al., 2024).

Another example of where variability in solar irradiance has an impact is in solar energy production, which primarily depends on total surface solar irradiance. The efficiency of photovoltaics (PV) depends on the material's temperature, and thus on whether an amount of irradiance is distributed constant or variable in time (or space), but also on the material's own specific wavelength response (Dirnberger et al., 2015; Lindsay et al., 2020). However, the largest impact of variable irradiance for solar energy is that historically, electricity grids have been designed instead for a constant and predictable energy supply. This is a practical challenge in the transition towards renewable energy that needs to be overcome (Yang et al., 2022), and depends in part on improved forecasts of SSI.

Accurate modelling, let alone forecasting, of SSI in the presence of clouds is difficult. Numerical weather prediction has a too coarse resolution to resolve individual clouds, with grid spacings in the order of 1 km. Cloud-resolving models like high-resolution large-eddy simulation (LES) typically cannot run on domains large enough to fully resolve the observed scale of atmospheric variability (e.g., van Stratum et al., 2023). Furthermore, radiative transfer calculations themselves are among the most computationally expensive components of models, and are therefore simplified. Common approaches are the independent-column approximation (1D instead of 3D radiative transfer), reduced spectral resolution, and reduced call frequency (Hogan and Bozzo, 2018). Because clouds and radiation are connected, errors in the representation of either component can affect the other. Studies exploring this connection beyond the aforementioned cumulus clouds are limited to offline calculations (e.g. Keshtgar et al., 2024).

Even without numerical constraints, there is significant uncertainty in the realism of the simulated state of the atmosphere, simply due to a lack of observations to initialize or validate models with. This largely motivated our observational approach in previous work, in which we analyse high-resolution SSI time series (Mol et al., 2023b) and gather and analyse new spatial observation of surface solar (spectral) irradiance (Mol et al., 2024). The development of a GPU-accelerated Monte Carlo ray tracer for accurate radiative transfer within the project this research is embedded (Veerman et al., 2022) now allows us to test and demonstrate insights gathered from the observational work.

Next, in Sections 2.1 and 2.2, we will more precisely define what we mean by *variability in surface solar irradiance* (SSI), demonstrate this with a diverse selection of (extreme) SSI variability from our observations, and discuss similar earlier studies by others. Then, in Section 2.3, we formulate our hypothesis that SSI variability can be explained by four mechanisms. These mechanisms are based on the observations discussed in Section 2.2 and further motivated by previous research on this topic.

From the selection of observational examples we design idealised numerical experiments, both to qualitatively reproduce the examples and to demonstrate how the four mechanisms work and when they are effective. Once the basics are established, we introduce more realistically simulated cloud fields to see how the hypothesis holds up when the complexity of clouds and cloud fields increases. The tools we use to simulate the cloud fields and calculate radiative transfer are introduced in Section 3,

followed by an overview of all experiments in Section 4. Results are presented and discussed in Section 5, and lastly, Section 6 concludes this study with an outlook.

## 2  Observed variability and proposed mechanisms

### 2.1  Definition and interpretation of surface solar irradiance variability

SSI variability at cloud-scale expresses itself in diverse ways, as will be shown shortly, and the choice of any definition is subjective. Here, we focus on conditions in which SSI exceeds clear-sky SSI, referred to as cloud-induced irradiance enhancement, or *irradiance enhancement* (IE). We further characterise SSI variability by having frequent and rapid fluctuations between shading (attenuated direct irradiance) and at least 10 % IE, spatiotemporal SSI patterns with IE and shading in excess of minutes or kilometres, or single, extreme maxima in SSI (IE > 30 %). This loose definition purposefully excludes some

common but less pronounced SSI variability conditions, because their impact is lower and we assume conditions are easier to understand when SSI variability is more pronounced. Locally enhanced diffuse irradiance and horizontally transported radiation are key features of cloud-driven SSI variability and naturally part of any condition of observed SSI variability that fits our definition.

Throughout this study, we will mostly present SSI normalised with, or relative to, clear-sky SSI to be able to compare across

cases with varying atmospheric conditions, surface albedos, and solar zenith angles. As small absolute changes in SSI can result in relatively large irradiance enhancement at high solar zenith angles, we only consider solar zenith angles of 75° or below.

### 2.2  Conditions of (extreme) surface solar irradiance variability

Figures 1 and 2 illustrate SSI variability in respectively time and space using high resolution observations. The time series are

based on the 1 Hz BSRN data of Mol et al. (2023b) or on 1-minute data of the Veenkampen station (https://maq-observations.nl/veenkampen/). The collection of spatial patterns is based on observed SSI measurement from a 50 m resolution spatial network at 1 Hz described in Mol et al. (2024).

### 2.2.1  Shallow cumulus

The shortwave radiative effects of shallow cumulus are the best described and studied among cloud types in the field of solar

irradiance variability research. Key features are the fast and frequent transitions between shade and sunshine (Figure 1a), and the bimodal distribution of SSI (e.g.,  Schmidt et al., 2009; Gristey et al., 2020; Tijhuis et al., 2023). In the distribution, one peak is centred around the typical value of diffuse SSI in cloud shadows, where no direct irradiance is able to reach, the other peak resembles the irradiance enhancement in directly sunlit areas. The shadows are typically dark (50 to 80 % lower than clear-sky SSI) and the transitions to sunlit areas in the order of tens of metres, illustrated by examples of spatial SSI patterns

of cumulus in Figure 2.

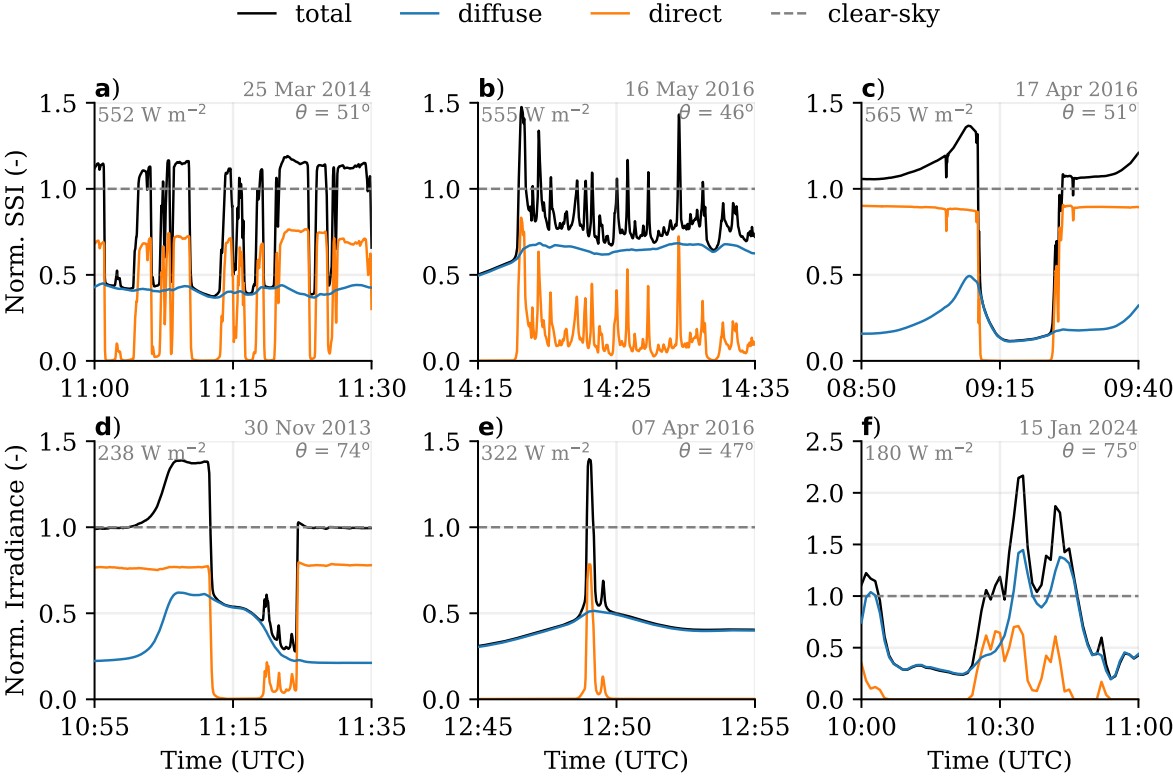

**Figure 1. A collection of observed time series featuring irradiance variability under different conditions.** The conditions are **(a)** shallow cumulus, **(b)** altocumulus, **(c)** cumulonimbus, **(d)** stratus field passage, **(e)** gap in stratocumulus, and **(f)** convective snow shower passage. The total, diffuse, and direct horizontal SSI are normalised with clear-sky values based on CAMS McClear (Gschwind et al., 2019). Average clear-sky SSI, solar zenith angle, and date are given in grey in the top of each subplot. Note that **(f)** has a larger y-axis.

### 2.2.2 Altocumulus

Altocumulus cloud fields result in a bimodal SSI distribution too, but increased cloud cover fraction reduces how frequently irradiance enhancement occurs. Diffuse irradiance is generally higher than seen in most other conditions of SSI variability and remains relatively constant on the spatiotemporal scale of individual irradiance enhancement peaks. Direct irradiance is often not completely attenuated, meaning the clouds are optically thin. Brief openings between individual altocumuli allow for direct irradiance to fully pass through and combine with the enhanced diffuse irradiance at the surface, thereby producing some of the most extreme observed IE peaks. Figure 1b shows all of these features in a time series, with one peak reaching nearly 50 % above clear-sky SSI. Fields of altocumuli were frequently observed while we did spatial SSI measurements during the FESSTVaL campaign (Mol et al., 2024). Figure 2 shows five different SSI patterns as a result of altocumulus, and one that includes cumulus as well. Peak irradiance enhancement varies between 30 and 60 % in these patterns. There are not many

studies on altocumulus, but at least Schade et al. (2007) and Yordanov et al. (2015) also identified specifically altocumulus as being highly potent in creating such significant irradiance enhancement.

### 2.2.3 Cumulonimbus

Deep convective clouds with a vertically developed structure and anvil at the tropopause have the largest area of influence of the cloud types we consider in this study. The sheer size and scattering surface area of a single cumulonimbus cloud make it an ideal candidate for casting long shadows and enhancing diffuse irradiance on the sunlit side. Segal and Davis (1992) identified some of these features in observations, in particular the long-lasting irradiance enhancement at the sunlit side of the cumulonimbus clouds. Figure 1c shows the passage of a relatively isolated cumulonimbus moving from north to south over the sensor location with the Sun in the south, from our own observations. The key SSI pattern features for this cloud-type are the slow ramping up of SSI on the sunlit side of the vertically developed cloud (until 09:10 UTC), subsequent rapid reduction to SSI values below even clear-sky diffuse irradiance, and the return to clear-sky SSI after the Sun reappears (09:25 UTC). In the observational example, this pattern is superimposed on a 5 % background enhancement of SSI that likely arises from other clouds in the vicinity. A fast rising cloud top of a cumulus congestus caused the shadow in the bottom right pattern of Figure 2, which is similar (but in reversed order) to the reappearance of sunshine in the time series pattern.

### 2.2.4 Stratus and stratocumulus

Stratus and stratocumulus are generally optically thick cloud types that result in little SSI variability, unless they dissolve (or form), have gaps in them, or otherwise advect into or away from clear-sky regions. Irradiance enhancement is not commonly associated with these cloud types, because the areas in which such variability can occur is small compared to the total cloud size, or brief compared to the total cloud lifetime. However, transitions from clear-sky to overcast conditions (and vice versa), or cloud gaps within overcast cloud fields, are associated with extreme irradiance peaks. Examples are respectively illustrated in Figures 1d and 1e. There is also a spatial SSI pattern caused by a small cumulus that formed in a cloud gap shown in the top left of Figure 2. Note that the gaps discussed here are larger and better defined than the smaller openings between altocumuli.

### 2.2.5 High albedo with broken cloud cover

A high albedo can enhance cloud-induced SSI variability by a significant amount, sometimes leading to extreme irradiance enhancement beyond what is normally observed under broken cloud cover. In Figure 1f, we show an example of a convective snow shower bringing a fresh layer of snow cover, which pushed the diffuse irradiance above clear-sky SSI values for multiple minutes. Such a measurement seems unrealistic, but has been observed before, see for example Gueymard (2017), their Figure 4. Villefranque et al. (2023) furthermore identified that surface albedo plays a significant role in SSI variability under cumulus clouds.

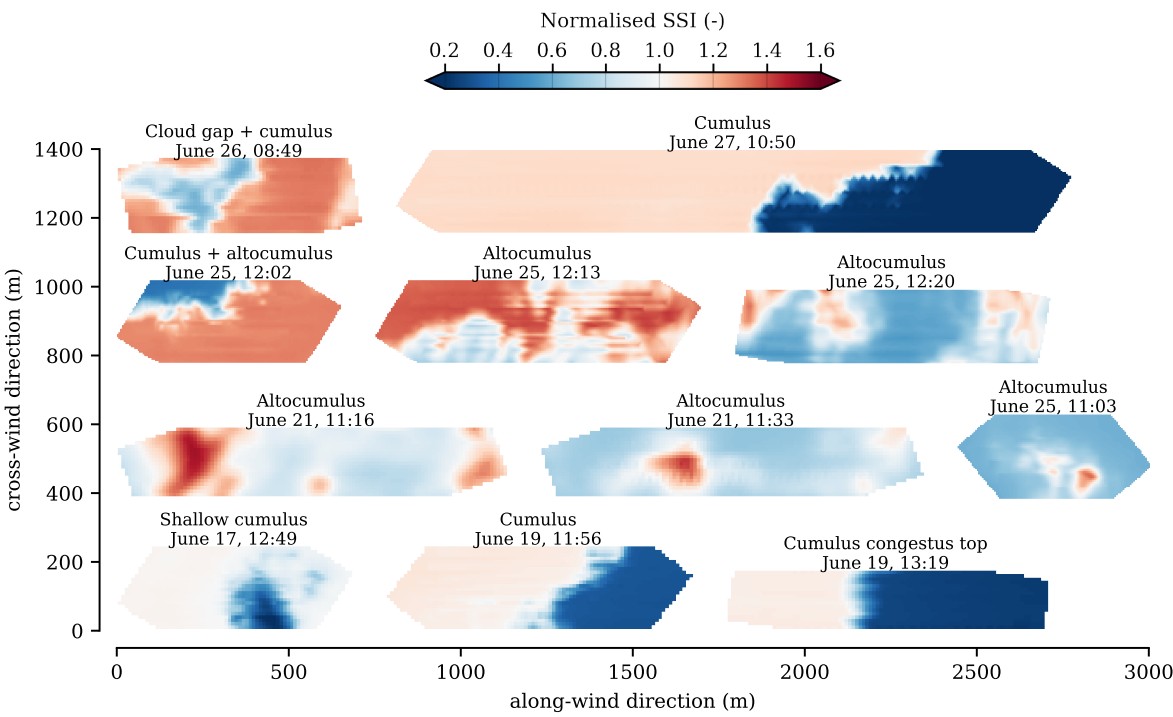

**Figure 2. A collection of observed spatial patterns of surface solar irradiance.** The patterns are spatially interpolated using temporal data as described in Mol et al. (2024) The labels indicate cloud type and date and time of occurrence (UTC), all in the year 2021. SSI is normalised with clear-sky SSI from McClear.

### 2.3 Proposed mechanisms

Cloud fields can manifest in a countless number of different configurations, each resulting in a unique surface irradiance field. In all cases, however, it starts with scattering of direct irradiance that is horizontally and diffusely redistributed onto the surface. We hypothesise that four key mechanisms by which solar radiation is horizontally redistributed can explain all the previously described prototype examples of SSI variability and irradiance enhancement peaks. Figure 3 schematically shows the four mechanisms. These mechanisms are in part based on prior research and may be known under different names, hence we will review and (re)define the terminology as we discuss each mechanism.

#### 2.3.1 Forward escape

Between transparent and opaque clouds, there is a range of optical thickness where direct irradiance is scattered mostly only once or twice, if at all. In the case of cloud droplets, we are in the regime of Mie scattering. Thus, 90-99 % of scattering occurs within 5° of the forward direction (calculated using `miepython`, Prahl, 2023), depending on droplet radius and photon wavelength. Consequently, part of the direct irradiance is scattered to an area on the underlying surface just besides the direct beam path, leading to irradiance enhancement if the direct path to this area is sufficiently cloud-free. This type of scattering occurs at cloud edges, where optical thickness gradually reduces to zero, and in other optically thin (parts of) clouds.

Small isolated cumulus clouds, for example, were found by Robinson (1977) to affect an area on the surface of approximately 5 to 15° off the direct beam centre. Yordanov (2015) produced IE of ∼ 80% in a simulation where an optically thin cloud was configured in a ring (cloud gap edge) with an opening the apparent size of the Sun's disk, although it is debatable whether such a configuration can realistically occur. The optical thickness resulting in the highest IE was found to be 3.1, similar to the optima found in simulation studies by Wen et al. (2001) and Pecenak et al. (2016). Although scattering by optically thin clouds is limited when expressed per unit cloud area, we expect this can become substantial for clouds or cloud fields with a significant amount of optically thin area. Altocumulus as shown in Figure 1b is an example of such an optically thin cloud field, indicated by the fact that direct irradiance is not fully attenuated.

#### 2.3.2 Downward escape

As cloud optical thickness increases beyond semi-transparent values, scattered radiation loses its dominant forward component due to multiple scattering. While radiation then increasingly gets scattered back up, the large remaining part that does not get absorbed by the cloud is highly diffusely transmitted downward through the cloud, thereby creating irradiance enhancements as part of it lands in adjacent sunlit areas. We call this mechanism *downward escape*, a term introduced by Várnai and Davies (1999), although their definition is expressed in terms of the horizontal photon transport bias in the 1D radiative transfer approximation.

*Downward escape* differs from *forward escape* in the location where the diffuse irradiance lands and how the diffuse irradiance coincides with direct to create extremes in SSI. Pecenak et al. (2016) show the transition between *forward escape* and *downward escape* clearly in 2D simulation experiments of rectangular clouds, with the peak of enhanced irradiance moving

from near the projected shadow location to underneath the cloud as optical thickness increases. Optically thick but flat clouds, like the stratus and stratocumulus of Figure 1, are likely candidates for where this mechanism is a dominant factor in creation of SSI variability. Solar zenith angle will play an important role in the effectiveness of *downward escape* as a mechanism, as it determines to what extent direct irradiance can coincide with the peak in diffuse enhancement near or underneath the cloud.

### 2.3.3 Side escape

Optically thick clouds scatter a large part of the solar irradiance back to space. Similarly, under solar zenith angles higher than $0°$, the sides of vertically structured clouds have the same effect, except due to the vertical orientation the scattering is partially directed towards the surface. Segal and Davis (1992) described these effects of deep convective clouds as reflections, but Várnai and Davies (1999) refer to this phenomenon more accurately as backscattering. We will call it *side escape*, where cloud sides act as a region where photons diffusely escape after multiple scattering events, rather than reflecting radiation like a mirror.

*Side escape* is also discussed in the context of biases in the independent-column approximation, where photons that enter a cloud from the top can not escape from its sides (e.g. O'Hirok and Gautier, 1998). We expect, however, that the relative importance of *side escape* scales with the amount of radiation entering the side of the cloud, as photons are more likely to exit where they enter instead of travelling a long distance through the cloud. We also expect this mechanism to become effective as the total vertically oriented surface area increases compared to the horizontal size of the cloud, typical in cases of deep moist convection. In such cases, optical thickness will generally be high and thus *forward escape* will be ineffective and most radiation escapes out of the closest edge, i.e., the sunlit side. We thus also expect strongly asymmetric surface patterns of irradiance with vertically structured clouds due to diffuse irradiance being enhanced primarily on the sunlit side, as we find in the example in Figure 1c.

### 2.3.4 Albedo enhancement

Surface albedo can enhance the cloud-enhanced surface irradiance by multiple iterations of scattering between surface and cloud base. This mechanism may explain the extreme irradiance peaks observed during snow cover (high albedo) combined with broken cloud cover as reported by Gueymard (2017), or shown in Figure 1f. In a more general sense, this mechanism is known as *entrapment*, which can occur between any two scattering surfaces (Schäfer et al., 2016; Hogan et al., 2019). In this study, we focus on the entrapment of radiation between a surface (land or ocean) and a cloud (field). The effectiveness of this mechanism, aside from higher surface albedo, will increase with cloud cover and optical thickness, required to scatter back reflected surface irradiance, up to the point where too little irradiance is available. Villefranque et al. (2023) have shown this effect for uniform clouds by varying optical thickness and albedo and found similar behaviour for cumulus clouds. We therefore expect areas at the edge of stratus, gaps in stratocumulus, and perhaps large cumulus clouds are candidates for significant *albedo enhancement*.

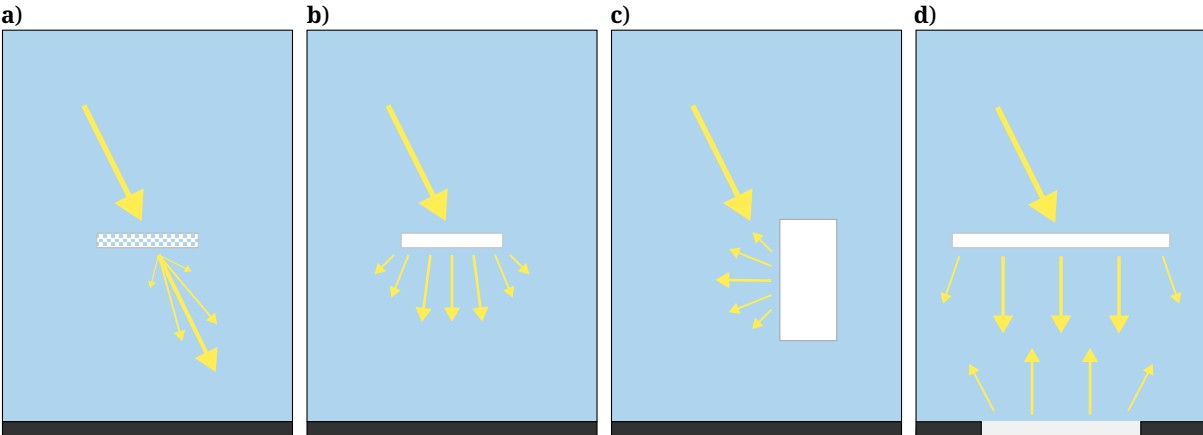

**Figure 3. Schematic representation of the four proposed mechanisms that drive surface solar irradiance variability**. From **(a)** to **(d)**, these are *forward escape*, *downward escape*, *side escape*, and *albedo enhancement*. Incident solar radiation is indicated by the large downward arrow, the other arrows indicate approximated average direction and intensity of scattered radiation. Surface albedo is indicated by the lightness of the surface. Only the arrows relevant to each mechanism are drawn.

## 3    Simulation tools

We use a Monte Carlo ray tracer (MCRT) to simulate 3D radiative transfer within cloud fields overlying a surface. With a ray tracer, we can resolve the complex 3D paths individual photons take through a cloudy atmosphere overlying a surface, instead of the simplified but much faster 1D radiative transfer solvers. The cloud fields either come from case studies run using large-eddy simulation (LES) where we simulate specific cloud types, or they are manually created. The combination of the MCRT with these cloud fields allows us to isolate, to some extent, the contribution of the four mechanisms to the total surface irradiance variability. Information on creating the cloud fields and the ray tracer setup in general is given below. The details of each experiment are demonstrated in the next section. All model code, input data, and case setups are freely available, see our data availability statement at the end.

### 3.1    3D radiative transfer

For simulating 3D radiative transfer, we use the GPU-accelerated version of the Monte Carlo ray tracer introduced by Veerman et al. (2022), in single-precision mode. The ray tracer is a flux solver built on top of RRTMGP, the optical solver that is part of RTE+RRTMGP (Pincus et al., 2019), and is fast enough to run coupled to the LES, although in this study we use it in offline mode. One of the optimisations is a reduced spectral resolution, via smaller sets of so-called g-points, chosen in such a way as to have minimal impact on accuracy (Veerman et al., 2024). We use the 112 set for shortwave radiation, halving the calculation time. This time is reinvested in spatial resolution, number of rays, and the use look-up tables for an accurate Mie scattering phase function. A Mie scattering phase function is important to resolve the narrow but dominant forward peak in scattering direction in the case of cloud particles. For simulations that include ice particles, we use a Henyey-Greenstein phase function,

which has a more diffuse forward peak, as our Mie look-up table for ice particles is not yet validated. The effects of a different phase function are minor for our experiments that contain ice, as these are on relatively large and coarser resolution domains (see sensitivity analyses in Appendix A1). For all simulations, we use the same background (trace) gas profiles, except for water vapour. Lateral boundary conditions for the MCRT are periodic, therefore some simulation domains need to be made

large enough to isolate the effects of a single cloud. As for the number of rays, or sample per pixel, we set this to a high enough to get a clear signal, typically 256 to 1024.

We sample optical thickness $\tau$ from the MCRT, which gives a 3D field of $\tau$ per grid cell and wavelength band. We choose a band in the relatively energetic visible spectrum (band 10, corresponding to 625 to 768 nm) when discussing values of $\tau$ in the experiments, for simplicity. $\tau$ will be typically higher for longer wavelengths and vice versa, but the differences are small in

the context of our study. For example, $\tau$ for band 8 (1242 to 1298 nm) is 3 to 6 % larger than band 12 (345 to 442 nm) in our experiments.

For a given amount of condensate, $\tau$ also varies with the effective droplet radius, which in turn depends on the droplet number concentration. We set the droplet number concentration for liquid water and ice constant at respectively $10^8$ m$^{-3}$ and $10^5$ m$^{-3}$ for all cases. The effective radius for liquid water droplets is bound between 2.5 and 21.5 μm, or between 5 and 90 μm

for ice particles. As a result, an increase in $\tau$ due to increased total liquid (or ice) mixing ratio is slightly offset by an increased effective radius. Effective radii have a noticeable effect on calculated SSI fields, primarily by changing $\tau$ given an amount of condensate (see sensitivity analyses in Appendix A2). Results are therefore mostly interpreted with respect to $\tau$, making our conclusions robust with respect to our choice of number concentration.

For some cases, we solve radiative transfer online (i.e., coupled to the model) in order to simulate more realistic clouds

that are radiation driven, such as altocumulus. We then use the standard RTE+RRTMGP solver with the independent-column approximation, which is much faster and good enough to create the cloud fields we want. Any analysis or illustrated data of irradiance we demonstrate will come exclusively from the 3D MCRT.

## 3.2 Simulated and synthetic cloud fields

We use MicroHH in LES-mode as our cloud-resolving atmospheric model (van Heerwaarden et al., 2017). The model is

235 initialised with vertical profiles of temperature, moisture, and wind, either from observed soundings, reanalysis data, or from idealised experiments based on literature. As a consequence, clear-sky SSI varies between some simulations, but results are generally normalised against their respective clear-sky values to provide a more general interpretation. The lateral boundaries are periodic, and domain size and spatial resolution are case-dependant. For the synthetic cloud fields, we start by initialising the model and then manually modify the cloud fields, either by changing the temperature in a layer and letting the model

condense water vapour, or by directly modifying the liquid ($q_l$) or ice ($q_i$) fields.

We can manually move any cloud field up and down in altitude to test effects of cloud height, or test a range of optical thickness by increasing or decreasing the amount of condensate while keeping the cloud geometry and location fixed. The surface is flat in all simulations. Albedo is spatially homogeneous, wavelength-independent, and kept the same for direct and

diffuse radiation. In some simulations we prescribe homogeneous surface fluxes or run with online radiation and an interactive surface layer.

## 4 Experiments

What now follows is an overview of all cloud fields that we create. Experiments start idealised and become progressively more complex and detailed. We run the 3D radiative transfer model on these cloud fields and vary a set of parameters across a range of values, depending on the specifics of the experiment. These parameters are surface albedo $\alpha$, solar zenith angle $\theta$, solar azimuth angle $\phi$, cloud optical thickness $\tau$, cloud altitude $h$, and cloud depth $d$. All simulations are run with a surface albedo of 0, unless stated otherwise. For all configurations we run the radiative transfer model also once without clouds to estimate clear-sky irradiance ($\mathrm{SSI_{cs}}$).

### 4.1 Synthetic cloud fields

#### 4.1.1 Flat clouds

The horizontal geometry of four synthetic clouds fields are shown in Figure 4. The `stratus` case cloud covers half the x-domain, is 3 grid-points thick (150 m), and sits at an altitude centred around 450 m above ground level. The (x, y, z) domain size is $12.8 \times 3.2 \times 3.2 \ \mathrm{km}^3$ ($\Delta$ x,y,z = 50 m), the smallest possible domain size in which the centre of the cloud-free part (x = 0 m) is far enough away that its surface irradiance approaches that of clear-sky conditions. Due to periodic lateral boundary conditions of the radiative transfer solvers, the y-domain is effectively infinite as long as the solar azimuth angle is kept exactly aligned to the y-direction. Vertically integrated liquid water is 0.158 kg m$^{-2}$ ($\tau \sim 19$) and is generated by manually cooling a thin layer in an observed atmospheric profile is that is otherwise relatively dry in the lower 3 km (see *Code and data availability*). This is an idealisation of a stratus to clear-sky transition (and vice versa), such as the example in Figure 1d.

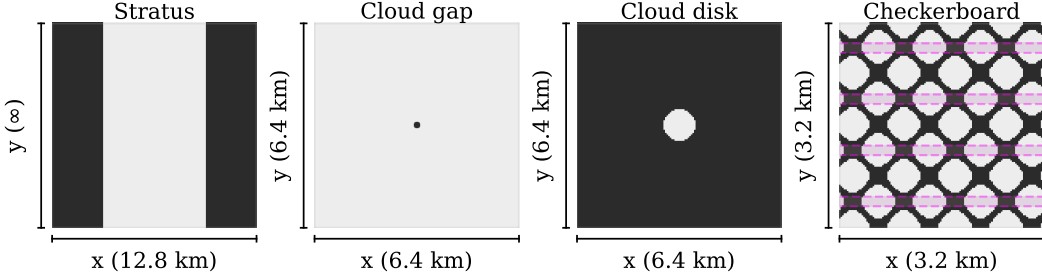

**Figure 4. Top view of the synthetic cloud fields.** Light greys indicate clouds, dark grays are the cloud-free areas. These clouds are thin relative to their horizontal size (cloud depth varies between 25 m and 150 m). Cloud depth, altitude, and optical thickness depend on the case and experiment. Note that the horizontal domains vary in size per case. The magenta shading highlights the area selection used in Figure 9.

For the `cloud gap`, we use the same atmospheric profile and general configuration as in the `stratus` case, but now the synthetically created stratus stretches across the whole domain and has a circular gap in its domain centre with a 100 m radius. The gap size was chosen to be small yet still resolved given the horizontal resolution (which we increased to 25 m for this reason), and while still letting through direct sunlight under non-zero solar zenith angles. The (x, y, z) domain size is changed to $6.4 \times 6.4 \times 3.2$ km$^3$, sufficiently large to have no radiation effects from the periodic boundary conditions. Results are averaged in the cloud gap across a 150 m subset centred in the y-direction In our experiment, the gap is a factor 50 wider than the apparent diameter of the Sun (200 m compared to $\tan(0.5^{\circ}) \times 450$ m $\approx 4$ m). This resembles the stratocumulus with a gap shown in Figure 1e, rather than a gap between optically thin altocumuli as simulated by Yordanov (2015).

The `cloud disk` case is conceptually the inverse of `cloud gap`, as can be seen in Figure 4, but only has a thickness of 1 grid-point (50 m). That is unrealistically thin, but necessary to minimize the effect of *side escape* in experiments where we manually vary its total condensate to control optical thickness. Cloud disk altitude and diameter are also varied in experiments, and therefore the vertical extent of the domain is increased to 9.6 km. Horizontal resolution is also 25 m to resolve the circular shape. This cloud is for testing the effect of a singular patch of optically thin cloud, which may resemble a small cumulus, a transparent cloud edge, a single patch of altocumulus or a piece of cirrus.

The `checkerboard` case is a different setup and features a repeating checkerboard-like pattern of 100 m thick patches that are 500 m in diameter, with at most a 250 m spacing in between patches, and sits centred around an altitude of 2850 m above ground level. Cloud optical thickness is $\tau \approx 2$. The (x, y, z) domain size is $3.2 \times 3.2 \times 4.8$ km$^3$ ($\Delta$ x,y,z = 25 m). This idealisation is inspired by an observed case of altocumulus in FESSTVaL (see Section 4.2), with similar properties as shown in Figure 1b. The atmospheric profile used to initialize this cloud field is therefore also different from the previously described cases, and will have clear-sky SSI values that are $\sim 5$ % lower for a given solar zenith angle.

### 4.1.2 Vertical clouds

For studying the effect of cloud sides in the most simple way, we take the `cloud disk` and extend it vertically, effectively creating a perfectly homogeneous and smooth cylinder. The cloud base is at 1000 m above ground level and cloud top varies from 1500 to 12000 m, and therefore the cloud depth varies from 500 to 11000 m. Total liquid water path and cloud optical thickness $\tau$ increase linearly with cloud depth as we copy existing cloudy grid cells without rescaling the liquid water content per grid cell.

Due to the large area of influence of the clouds in this experiment and their deep vertical extent, we have to increase the (x, y, z) domain size to $76.8 \times 76.8 \times 19.2$ km$^3$. As a consequence, resolution was reduced to 100 m in the vertical and 300 m in the horizontal. This case is an idealisation of a growing deep convective cloud (cumulus congestus) in an environment free of wind shear, and therefore called `towering cumulus`. Figure 1c shows the passage of such a deeply developed cloud, although there is also an anvil cloud present in this observation and a real cloud is naturally influenced by turbulence and thus not smooth.

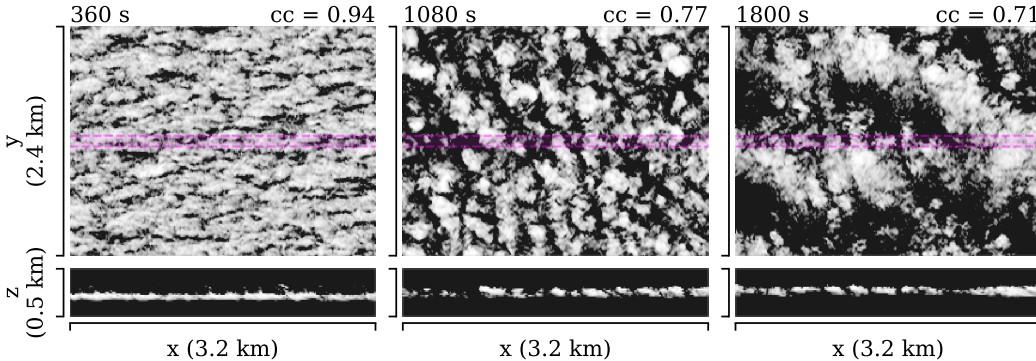

**Figure 5. Simulated altocumulus evolution**. The top row shows a top view of vertically integrated liquid water ($q_l$), the bottom row shows a side view of horizontally integrated $q_l$ over a 200 m slice in the y-direction (marked by the magenta shading). Cloud altitude is $\sim$ 2800 m. The full domain is $6.4 \times 6.4$ km$^2$, only a subset is shown. Time step and cloud cover are labelled at the top.

## 4.2 Simulated altocumulus

In this case, we simulate an altocumulus cloud field observed during the FESSTVaL campaign on June 21 (shown in Mol et al. (2024) or in parts of Figure 2). The model is initialised from a radiosonde at the observatory, which has a small layer of nearly condensed air at approximately 2850 m and drier air above and below that layer. By increasing the relative humidity beyond 100 % in this layer at the first time step, we force the creation of a thin layer of condensation. With small noise in the vertical velocity field at initialisation, the model runs freely with coupled 1D radiative transfer, which develops the layer of condensation into a thin field of dynamic altocumulus, as illustrated in Figure 5. Altocumulus requires a high simulation resolution in order to resolve the altocumulus, as the vertical depth and horizontal cell structures are small. Domain size is therefore on the small side for an LES, with $6.4 \times 6.4$ km$^2$ ($\Delta$x, $\Delta$y = 12.5 m) and a vertical domain of 4.2 km ($\Delta$z = 16 m). After 1800 s, we get unrealistic wave growth in our periodic domain, which explains the large structures that emerge at this time step.

## 4.3 Simulated cumulonimbus

To make the `towering cumulus` experiment more realistic we simulate two versions of an isolated cumulonimbus in our LES model. These are adaptations of the idealised supercell simulation setup introduced by Weisman and Klemp (1982). In one simulation, we disable vertical wind shear, in the other we set it to 25 m s$^{-1}$, resulting in respectively a straight vertical cloud and a tilted one. The horizontal domain is $153.6 \times 153.6$ km$^2$ ($\Delta$x, $\Delta$y = 200 m), with the domain top at 19.2 km ($\Delta$z = 150 m). A large domain is necessary to keep the cumulonimbus from scattering radiation onto itself due to periodic boundary conditions in the MCRT. Vertically and horizontally integrated liquid and ice water mixing ratios are qualitatively shown in Figure 6 for three time steps during the growing stages.

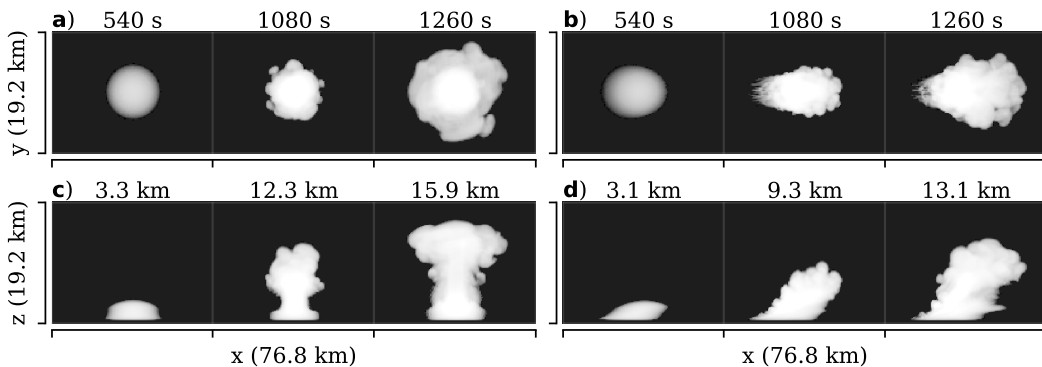

**Figure 6. Two simulated deep convective updrafts**. **(a, c)** show the updraft in shear-free conditions, **(b, d)** show the same updraft but with 25 m s$^{-1}$ vertical wind shear. White colours indicate high values of liquid and ice water mixing ratios ($q_l + q_i$) integrated along the z or y-axis. The time and cloud depth of each snapshot is labelled at respectively the top and bottom rows.

## 4.4 Simulated free convection

With the final simulation, called `free convection`, we aim to demonstrate what happens in a cloud field that is on the high-end of possible complexity, in contrast to the other experiments in this study. This simulation has a horizontal domain of 102.4 × 102.4 km$^2$ ($\Delta$x, $\Delta$y = 200 m), with the domain top at 19.2 km ($\Delta$z = 100 m). We initialise the simulation using an observed conditionally unstable thermodynamic profile with low wind shear. We include online 1D radiative transfer to heat up an interactive land surface which will thermodynamically initiate deep convection. Figure 7 shows this evolution of `free`

`convection` in three stages, from cumulus towards deep convection. After one hour, as the convective inhibition diminishes with daytime heating, widespread cumulus and cumulus congestus forms randomly, as there is no other forcing or form of organisation. In subsequent hours, convection consolidates in a smaller number of strong deep convective clouds with large anvils.

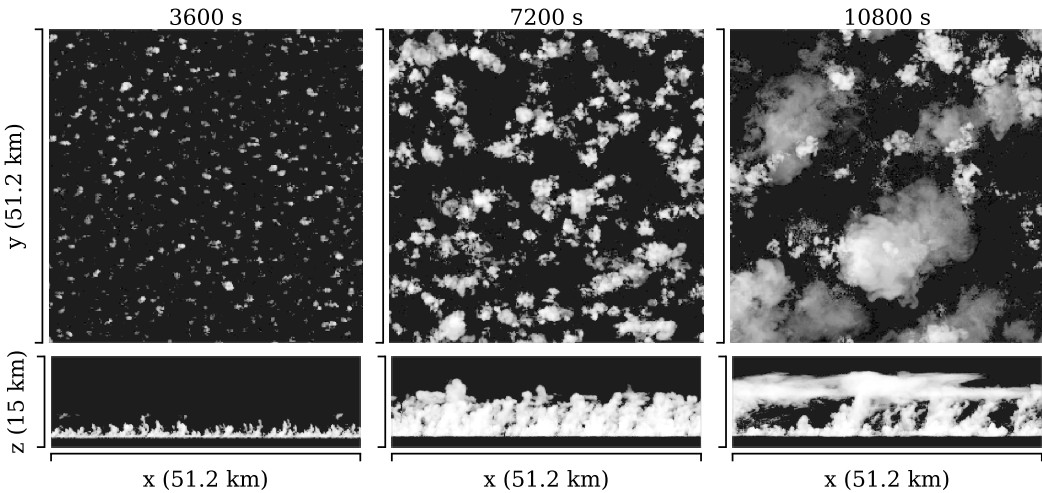

**Figure 7. Simulated free convection**. The top row shows the vertically integrated liquid and ice content ($q_l + q_i$), the bottom row shows the same integral along the y-axis. Only a quarter of the full horizontal domain is shown.

# 5 Results

## 5.1 Cloud optical thickness controls which mechanism dominates

We will first demonstrate the relationship between the mechanisms of *forward escape*, *downward escape*, and *albedo enhancement*. The interplay between these three mechanisms using the `cloud disk` case is illustrated in Figure 8 for a zenith angle of 45°.

For low optical thickness ($\tau \approx 0.2$), radiation is primarily scattered into the projected cloud shadow at the surface. At high optical thickness ($\tau \approx 44.2$), the forward projection disappears, and instead a wide area centred underneath the cloud is diffusely enhanced. Albedo has no effect on the low end of the optical depth range, whereas it contributes significantly at the high end ($\sim 4$ times more than *downward escape* at $\alpha = 0.8$). Intermediate values of $\tau$ show a transition from predominantly *forward escape* without *albedo enhancement* to *downward escape* with *albedo enhancement*.

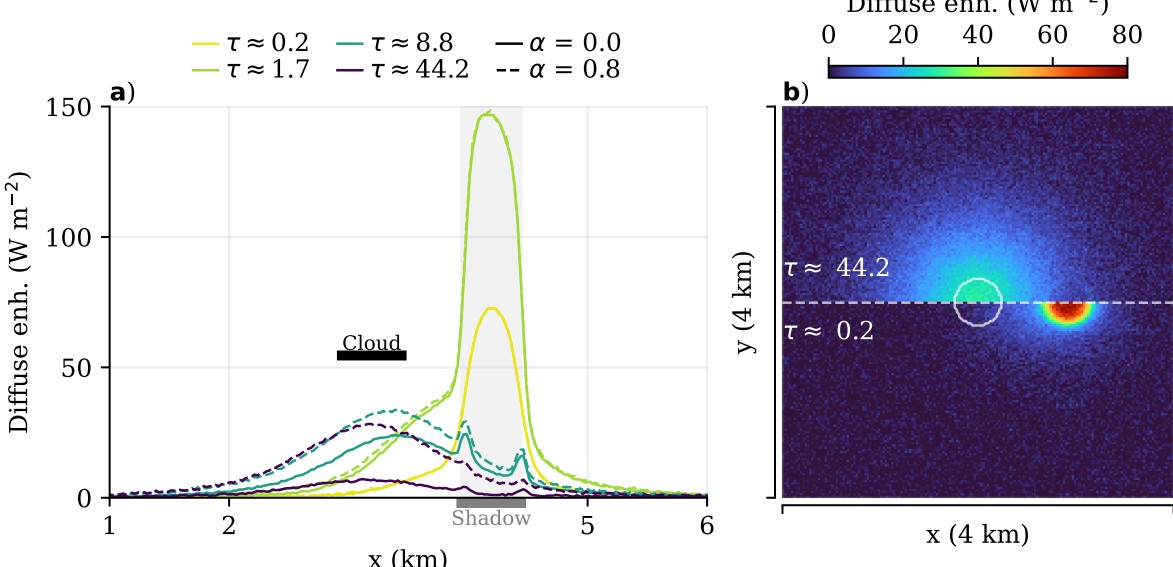

**Figure 8. From *forward escape* to *downward escape*.** A cloud disk at 1 km altitude with 25 m in depth and a 500 m diameter scatters radiation coming in at a zenith angle of 45°. Diffuse SSI enhancement is calculated relative to clear-sky diffuse values. The lines in **(a)** are averaged over the middle 400 m in the y-direction, whereas **(b)** shows the 2D surface field. The circle in the centre of **(b)** marks the edge of the cloud disk.

For this experiment, we can estimate a value for $\tau$ where the transition occurs as the point where the diffuse enhancement underneath the cloud exceeds that of within the projected shadow location. Estimated from Figure 8, this transition occurs between $\tau = 1.7$ and 8.8. Numerically, by simulating the values of $\tau$ in this transition range, we find *downward escape* takes over at $\tau > 6.3$, or $\tau > 5.4$ when *albedo enhancement* is included ($\alpha = 0.8$). Below $\tau = 1.7$, the effect of albedo becomes

indistinguishable. We keep these numbers for $\tau$ in mind for the next sections as an estimate for which scattering mechanisms are potentially at play.

What this experiment illustrates is that when *forward escape* dominates in a cloud field, *albedo enhancement* is negligible. Furthermore, for a uniform cloud, *downward escape* and *forward escape* only co-occur in the intermediate range of optical thickness. In more complex clouds or cloud fields, in which low and high optical thickness are common and close together, these mechanisms may still co-occur.

### 5.2   *Forward escape* and *downward escape* in simple, flat cloud fields

We will now further describe the *forward escape* and *downward escape* mechanisms using the simple, flat, synthetic cloud field experiments: `stratus`, `cloud gap`, and `checkerboard`. In Figure 9a-c, we show the resulting SSI patterns for these experiments at solar zenith angles of 0, 30, and 60°. All three resemble the patterns of diffuse, direct, and total SSI from the observations in Figure 1b, d, and e, respectively. Albedo effects are ignored for the time being, and note that the solar azimuth angle is 270° (i.e., from the west, or left).

Given the optical thickness regimes we estimated in the previous section, the patterns for `stratus` and `cloud gap` are driven exclusively by *downward escape* (in the absence of albedo). Diffuse irradiance is maximally enhanced underneath complete cloud cover, where also direct irradiance is completely attenuated. The largest irradiance enhancement above clear-sky in the `stratus` experiment is thus found at the transition where direct irradiance appears, but the effects of *downward escape* are still present (Figure 9a,d,g). Increasing the zenith angle in the `stratus` case shifts the direct irradiance underneath

the cloud on the sunlit side where diffuse irradiance is enhanced the more, which further increases the irradiance enhancement in this configuration its potential maximum.

    In contrast, for a 200 m diameter cloud gap the diffuse irradiance is almost uniformly enhanced below the cloud field. Any direct irradiance passing through the gap would coincide with the highest possible enhanced diffuse in the domain, independent of solar zenith angle, (Figure 9b). Widening the cloud-free gap reduces the diffuse enhancement underneath that gap, which

will make the case more similar to the previously described `stratus`. For a gap diameter that is close to the cloud vertical depth, direct irradiance will be unable to pass through and create any irradiance enhancement at higher solar zenith angles, as is the case for $\theta = 60°$ in Figure 9h. In an experiment like this, gaps with sizes that approach the apparent Sun diameter (0.5°) can rarely cause irradiance enhancement unless the zenith angle is 0° or the cloud is only meters thin.

    The `checkerboard` case is different, as the cloud field is optically thin, which results in a pattern that is dominated by

365 *forward escape*, still ignoring albedo effects for now. Diffuse irradiance is enhanced almost uniformly underneath the cloud field despite dominant *forward escape*. This is due to the high altitude of the cloud (2850 m), which allows the forward diffuse peaks of an individual cloud patch to be spread over a large enough surface area to blend together with the scattered irradiance caused by other patches. Peak diffuse still occurs in the projected shadow location where direct irradiance is partially attenuated. As a result, irradiance enhancement is approximately 45 % in the sunlit gaps between the cloud patches. This is similar to the

370 type of pattern and intensity of the irradiance enhancement as in Figure 1b, except the real-world geometry of altocumuli is more variable.

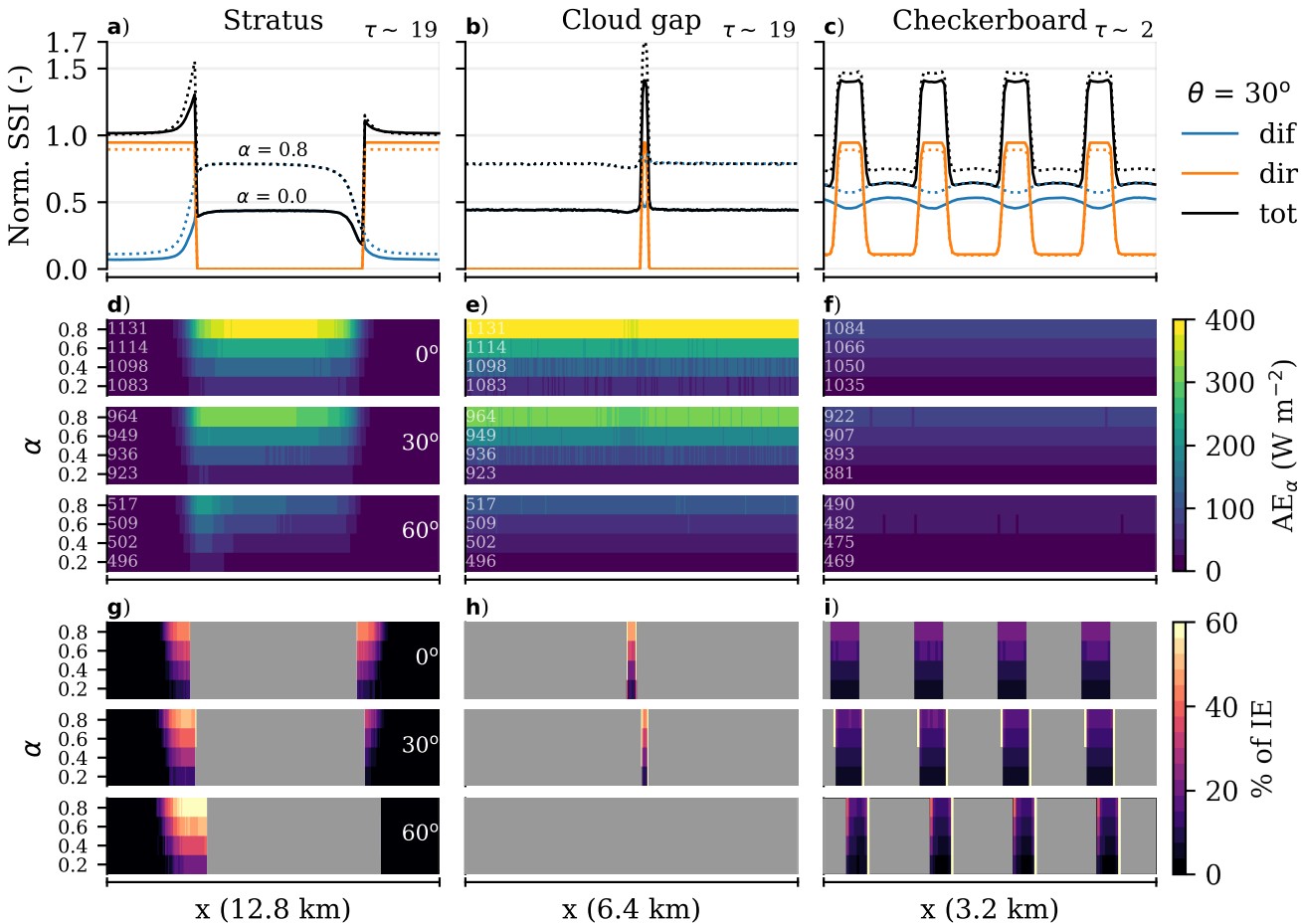

**Figure 9. Surface patterns for the stratus, cloud gap, and checkerboard cloud fields.** The top row shows the diffuse, direct, and total surface solar irradiance along the x-axis and averaged over the y-axis. Averaging over the y-axis is done over a subset of the checkerboard (magenta shading in Figure 4d) and the cloud gap cases. The middle row shows *albedo enhancement* for a given albedo ($AE_\alpha$) as the irradiance enhancement relative to $\alpha = 0$ for $\alpha \in [0.2, 0.8]$. The bottom row shows the relative contribution of $AE_\alpha$ to the total enhancement of irradiance $IE_\alpha$. Solar zenith angle is $30°$ for the top row, or 0, 30, and $60°$ for the other subplots. Clear-sky total SSI in W m$^{-2}$ is added for each solar zenith angle, albedo, and case in subplots d, e, and f

.

## 5.3 Albedo significantly enhances SSI under optically thick clouds

We will now introduce the *albedo enhancement* mechanism in these cloud fields. We quantify the *albedo enhancement* (AE) in the presence of a cloud for a specific surface albedo ($\alpha$) as $AE_\alpha = IE_\alpha - IE_{\alpha_0}$. Here, $IE_\alpha$ is the cloud-enhanced surface

irradiance for a given albedo $\alpha$, i.e., $\mathrm{SSI}_\alpha - \mathrm{SSI}_{\alpha,cs}$. $\alpha_0$ is the reference albedo of 0, representing the cloud-enhanced irradiance without any *albedo enhancement*. $\mathrm{AE}_\alpha$ is shown in Figure 9 for the `stratus`, `cloud gap`, and `checkerboard` cases.

The simplest effect can be seen in the patterns in Figure 9a-c, where the dotted lines are $\alpha = 0.8$, which all show a significant increase in diffuse irradiance and minor reduction in direct irradiance (normalised by their respective clear-sky values). The high cloud cover and optical thickness of the stratus cloud in the `stratus` and `cloud gap` cases result in significant *albedo enhancement*. Albedo enhances the diffuse irradiance most where local cloud cover is highest, which explains the patterns in Figure 9d and e. Again, low cloud optical thickness renders *albedo enhancement* negligible, this time shown by the weak diffuse enhancement in the `checkerboard` case (even for $\alpha = 0.8$).

Figures 9g-i show the relative contribution of *albedo enhancement* to the total IE, with shaded areas greyed out. This shows that at a modest albedo of 0.2, the albedo accounts for 10 % of the total IE in the stratus and cloud gap cases. For high albedo, this can increase to as much as 60 % ($\alpha = 0.8$) for the `stratus` case at $\theta = 60°$. Under such conditions, direct irradiance lands further underneath the optically thick cloud on the sunlit side, and thereby adds a significant amount of radiation that gets scattered multiple times between the cloud and the surface.

In the `cloud gap` case, *albedo enhancement* is similar to stratus, except the contribution of direct irradiance is lower, limited by the gap size. Still, while the relative contribution of *albedo enhancement* to IE is approximately 10 to 45 %, the total IE is larger in this case, reaching 70 % for $\alpha = 0.8$ at $\theta = 30°$.

For optically thick clouds and high cloud cover, *albedo enhancement* can thus be significant, already contributing $\sim 10\,\%$ to the total IE at a modest albedo of 0.2. These effects amplify as albedo increases, and so albedo effects will often play a role in combination with *downward escape*. In contrast, the `checkerboard` case shows only a 10 to 15 % contribution of *albedo enhancement* to IE for the most extreme case of $\alpha = 0.8$. This is consistent with the low optical thickness of the cloud field and thus limited potential for multiple scattering. At lower albedo, the contribution of *albedo enhancement* is negligible, meaning the extremes in SSI in optically thin clouds are nearly exclusively driven by *forward escape*.

## 5.4 Forward escape enhances SSI for low clouds or high cloud area

As previously discussed in Section 5.1, *forward escape* results in scattered irradiance that closely follows the direct beam, of which a fraction lands next to the partially shaded area, creating irradiance enhancement. Hence, peak irradiance enhancement is low, as most scattered radiation falls within the projected shadow of the cloud (see Figure 8). We think this is why SSI is barely, or not at all, enhanced near the edges of cumulus clouds, as can be seen in the spatial patterns in Figure 2. The areas of low optical thickness in cumulus clouds, found at the edges or for the whole of the smallest shallow cumuli, are so small compared to the whole hemisphere from which radiation originates, that *forward escape* contributes little to irradiance enhancement or SSI variability. So how can *forward escape* still be effective in creating (extreme) SSI variability?

### 5.4.1 How forward escape effectiveness varies with cloud altitude and area

To further understand *forward escape*, we will study its relationship with cloud altitude and area. For this, we create `cloud disks` of 100 and 1000 m in diameter with an optical thickness of $\tau \approx 2$. Solar zenith angle is 0° (overhead Sun), surface

albedo is 0, to respectively minimize side effects of clouds and exclude the effect of albedo. Domain averaged total SSI does not change with cloud height, so any changes in the SSI patterns are due to differences in the redistribution of radiation with cloud height.

Figure 10 illustrates the effect of both cloud altitude and area. First, for the 1000 m diameter `cloud disk`, the peak diffuse enhancement or total irradiance enhancement is increased by at least a factor of 2 compared to the 100 m diameter one. Second, by lowering the altitude of the cloud disk, the forward scattered irradiance is spread out over a smaller area and thus peak enhancement in both diffuse and total irradiance increases. The total irradiance enhancement is always lower than diffuse enhancement, because for low optical thickness most scattered irradiance still falls within the projected shadow of the cloud.

A small area of optically thin cloud at low altitude is still not very effective, as the peak in irradiance enhancement is only able to exceed 5 % with respect to clear-sky at an altitude below 500 m. Conversely, the larger cloud disk already exceeds clear-sky SSI by 10 % at 6 km, and peaks at 25 % around 500 m altitude. The small decrease below 500 m is likely due to the cloud disk being so close to the surface that scattered radiation can not spread out horizontally enough to maximally combine diffuse and direct irradiance. To create more extreme SSI values, we can further increase the cloud area and place it at a higher altitude to maximise the combination of all scattered irradiance. Essentially, this is what we hypothesise happens in altocumulus, which we will now test.

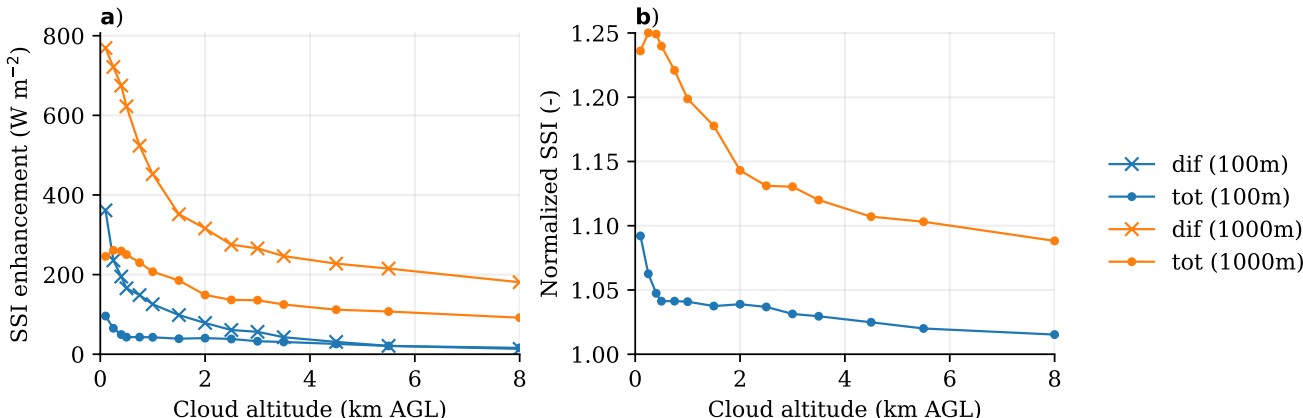

**Figure 10. Effects of *forward escape* with varying cloud altitude and cloud diameter.** Domain maxima for cloud diameters 100 m and 1000 m are shown here. In **(a)** the absolute surface solar irradiance (SSI) enhancement with respect to clear-sky (total or diffuse), and in **(b)** the total SSI enhancement normalised by clear-sky values. Solar zenith angle is $0°$, surface albedo is 0, and cloud optical thickness $\tau \approx 2$.

### 5.4.2 Why altocumulus fields create extreme SSI peaks

The simulated altocumulus fields affect the surface solar irradiance fields very similar to that of the `checkerboard` cloud field. Figure 11 illustrates the general patterns and dependence on solar zenith angle. Diffuse irradiance is relatively constant throughout the domain, with strong irradiance enhancements in the gaps between cloud patches. Higher total cloud cover (0.71

to 0.94), all of which has low optical depth ($\tau \sim 0.4$ to 0.6 on average), increases the total enhancement of diffuse and thereby increases the magnitude of the irradiance peaks. Higher cloud cover reduces the probability of these peaks occurring, however. Solar zenith angle affects the results primarily through reducing the effective size of the cloud-free areas, while increasing the relative magnitude of the enhancement.

The effect of albedo is also very small for the simulated altocumulus, despite high cloud cover. At an albedo of 0.8, the contribution to the irradiance enhancements is only $\sim 15\%$ (varying slightly with cloud cover and zenith angle). This means that the altocumulus field we simulate is nearly exclusively generating extreme irradiance variability by *forward escape*, even in high-albedo conditions. The 3D radiative effects should therefore be relatively simple to reproduce by a diffuse forward projection of the clouds combined with attenuated direct irradiance, and may not require an expensive model like we are using.

Our explanation of how altocumulus creates strong irradiance enhancement differs from that of Yordanov (2015), who emphasizes much more the local forward scattering at cloud edges surrounding a small gap near the apparent size of the Sun instead of a more uniformly enhanced diffuse irradiance. In that case, forward scattered irradiance that falls just outside the projected direct beam location would focus on one spot. Our simulated altocumulus field cannot resolve these very local effects, given that it requires resolved cloud edges at a resolution in the order of meters. The exact magnitude of such effects is hard to estimate, but in sensitivity experiments where we use a less strong forward scattering phase function (Henyey-Greenstein instead of Mie, Figures A1 and A2), SSI locally differs by roughly 5%. Additionally, we still find irradiance enhancement of the same magnitude as in observed cases, and instead believe that the contribution of meter-scale forward scattering effects to the total enhancement is small at best. We also deem a sharply defined cloud gap in the order of meters within a cloud layer with $\tau = 3.1$ (their optimal value) and a solar zenith angle of $0°$ unrealistic.

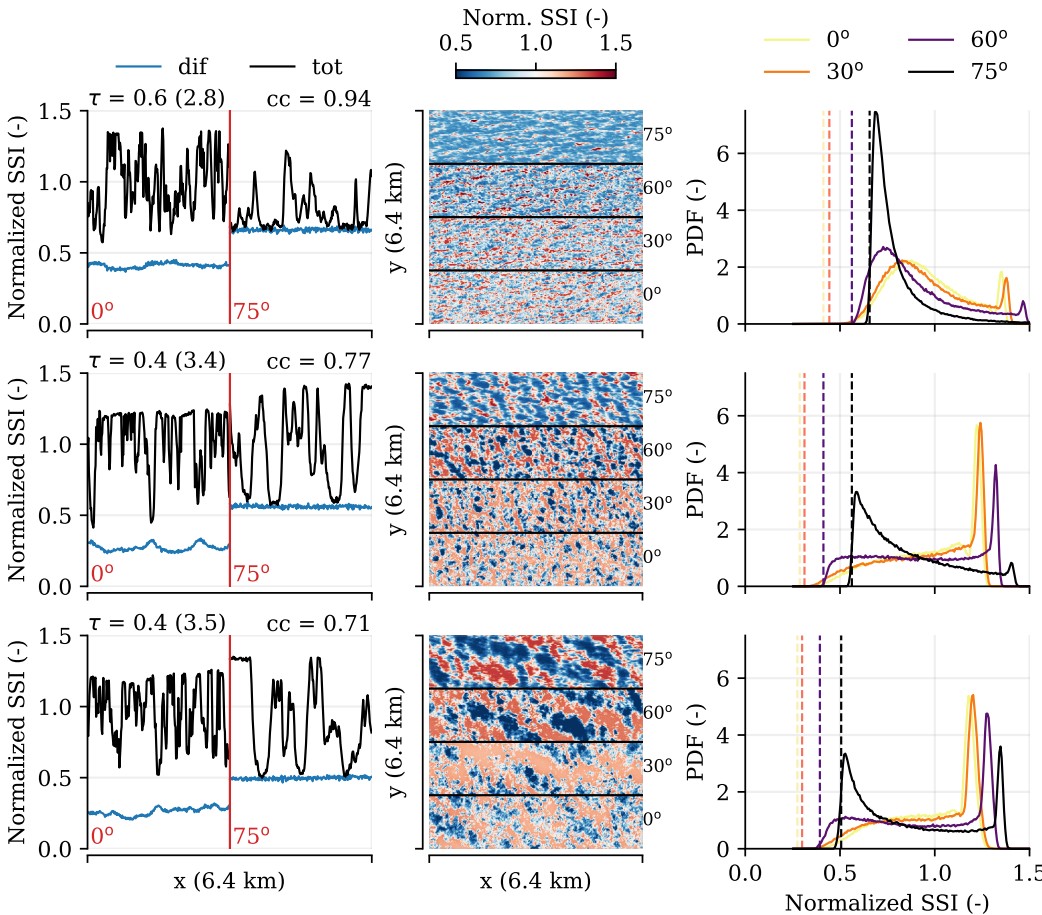

**Figure 11. Surface irradiance effects of simulated altocumulus.** The first column shows a section through the centre of the domain along the x-axis for a solar zenith angle of $0°$ and $75°$. The centre column shows for all simulated zenith angles the total surface irradiance field. The right column shows the probability density functions of these fields, with the value of the most probable diffuse irradiance added as dashed lines. The rows are individual time steps (360, 1080, and 1800 s), accompanied by decreasing cloud cover (see also Figure 5). Surface albedo is 0.

## 5.5 How cloud depth enhances SSI

We will now introduce the fourth mechanism, *side escape*, by analysing vertically developed clouds, starting with the synthetic towering cumulus and followed by two types of simulated and isolated cumulonimbus. For all experimental results that follow, the Sun is located in the west (azimuth angle of 270°), with a variable zenith angle $\theta$, and albedo set to 0. Furthermore,

integrated optical thickness is, for the most part, well beyond the *forward escape* regime in all experiments (typically $\tau > 50$), even when considering horizontally integrated optical thickness. There are two exceptions where *forward escape* still plays a significant role. The first is at non-zero zenith angles, where direct irradiance passes through the corners of an (optically thick) cloud. The second is for the edges of the simulated cumulonimbus, which are in or near the *forward escape* regime ($\tau < 10$), even when vertically integrating $\tau$. Before we further discuss these effects, we describe the general patterns of SSI for the

simpler case of `towering cumulus` and the role of cloud depth in these patterns.

### 5.5.1 Synthetic towering cumulus

A key feature of the SSI pattern in the presence of a vertically structured cloud is the significant irradiance enhancement on the sunlit side and (near) absence of enhancement on the shaded side. Figure 12a illustrates this for a `towering cumulus` 3 km in depth, 5 km diameter, and a 75° solar zenith angle. The SSI pattern is similar to the observed time series of a cumulonimbus

passage in Figure 1c, with a peak of enhanced diffuse irradiance on the sunlit side, a reduction underneath the cloud, and a return to (almost) clear-sky levels of SSI afterwards. As the `towering cumulus` increases in vertical depth, the rate of increase of the peak enhancement diminishes progressively, converging to a maximum of 15 % (for $\theta = 45°$, Figure 12b).

However, while the peak enhancement levels off, the area of influence extends further out, with IE still being a quarter of peak levels 10 km west of the sunlit side for the 11 km deep cloud. The total amount of diffuse enhancement in the domain

scales linearly with cloud depth ($r^2 \approx 0.99$, Figure 12c). Both the levelling off of the peak irradiance enhancement and increased horizontal extent of the surface pattern is explained by additional scattering occurring at increasingly higher altitude, which result in additional scattered diffuse radiation being spread out over a larger horizontal area on the surface.

Total scattered irradiance on the half of the domain that is on the sunlit side of the cloud, including increased top of domain outgoing radiation, is close to 100 % of the scattered direct irradiance intercepted by the cloud side. Scattering off of cloud

sides appears to be symmetrical in the vertical, at least for a perfectly straight and smooth cloud side, as half the radiation lands on the surface and half is radiated upwards out of the domain.

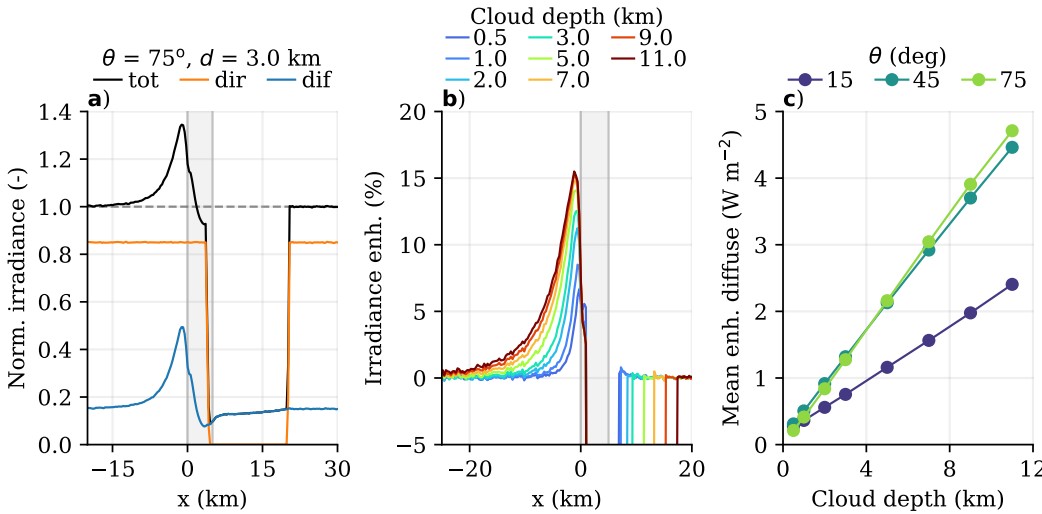

**Figure 12. SSI pattern for an idealised towering cumulus.** An example surface irradiance pattern is shown in **(a)**. The effect of increasing cloud depth on this pattern is shown in **(b)** for a solar zenith angle of $45°$. Mean diffuse enhancement on the sunlit side of the cloud is shown in **(c)** for varying cloud depths and zenith angles. Cloud base is at 1000 m, surface albedo is 0. The x-axis is relative to the location of the sunlit cloud side, where the cloud's position and width is illustrated by the gray shading.

### 5.5.2 Simulated cumulonimbus

In the cumulonimbus simulations, the updraft cloud is more turbulent and textured than the perfectly smooth `towering cumulus`. The SSI patterns remain largely the same, with significant enhancement only on the sunlit side close to the cloud and increased enhancement over a large area with increased cloud depth (Figure 13a). However, the peak irradiance enhancement in the shear-free case is $\sim 35\,\%$ and remains fairly constant with increased cloud depth, unlike the `towering cumulus`. Once the anvil cloud reaches far enough out to shade the whole updraft from direct irradiance, peak irradiance enhancement only slightly reduces from $\sim 35$ to $25\text{-}30\,\%$, visible in the last two simulation snapshots in Figure 13a (18.4 and 17.2 km).

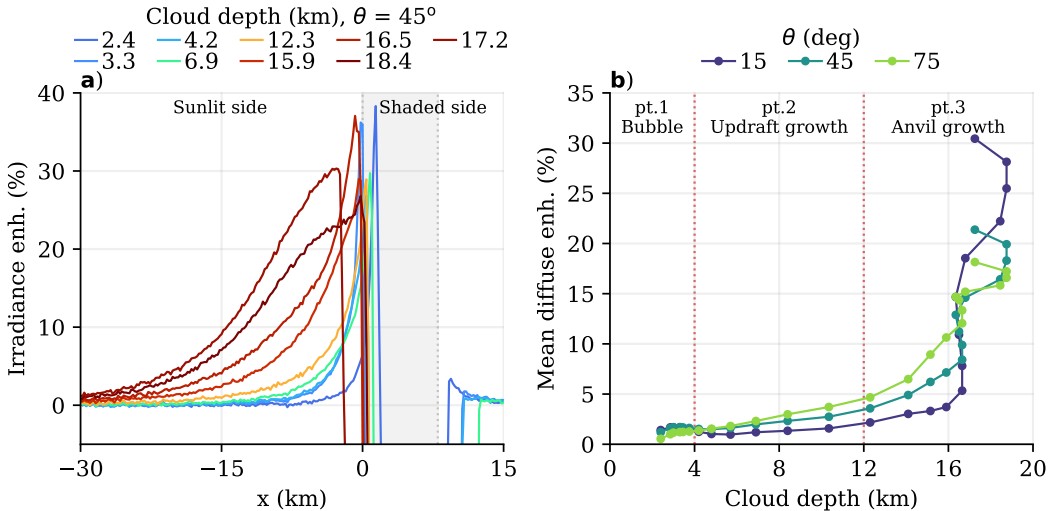

**Figure 13. Irradiance enhancement for a growing cumulonimbus in a shear-free environment.** The enhancement pattern in **(a)** is an average over the updraft width in the y-direction. The grey shading marks the approximate position and width of the updraft. Domain-mean surface diffuse enhancement on sunlit half is shown in **(b)** as function of cloud depth. Simulation evolution is marked by three phases (further explained in the text). Surface albedo is 0. The cloud top (and depth) decreases in the last time step.

The dynamic evolution of the simulated cumulonimbus also alters the relationship of total diffuse enhancement we observe in the domain. We identify three distinct parts, illustrated in Figure 13b. Initially, in part 1, the warm bubble that triggers the convection is wider than it is tall, and therefore diffuse enhancement is driven by *forward escape* and *downward escape*. During updraft growth, in part 2, the relationship between cloud depth and diffuse enhancement returns to the linear relationship observed for the `towering cumulus`. For higher zenith angles, diffuse irradiance is enhanced relatively more with cloud depth than for conditions with near overhead Sun ($\theta = 15°$). Finally, in part 3, the anvil growth takes over and accelerates the domain mean diffuse enhancement.

The disappearance of the narrow peak irradiance enhancement near the cloud edge due to anvil shading hints to *downward escape* and *forward escape* being the main mechanisms driving the irradiance extremes, rather than *side escape*. In support

of this, Figure 14 illustrates the relative position of the diffuse and total enhancement with respect to the cloud and direct irradiance path, with and without anvil shading at $\theta = 45°$. Without anvil shading, the peak diffuse enhancement lies directly underneath, and not in front of, the cloud edge at the sunlit side. Here, only *forward escape* and *downward escape* can occur, and thus these mechanisms drive peak enhancement. Most diffuse enhancement in front of the cloud comes from *side escape* before the anvil forms, as this scales with cloud depth. Once the anvil spreads out, it hinders the *side escape* mechanism and further enhances diffuse SSI by *forward escape* and *downward escape*, stretching out for $> 20$ km relative to the cloud centre.

The outer extent of the anvil is optically thin enough for several kilometres to let direct irradiance partly pass through and hit the lower kilometres of the updraft (depending on solar zenith angle). Modifying the optical thickness of the anvil thereby modulates how much *side escape* contributes to the overall SSI pattern. Lowering the optical thickness of the anvil leads to an overall brightening of the sunlit side of the cloud area (Figure A6 in the Appendix) and increase in *side escape*, and vice versa.

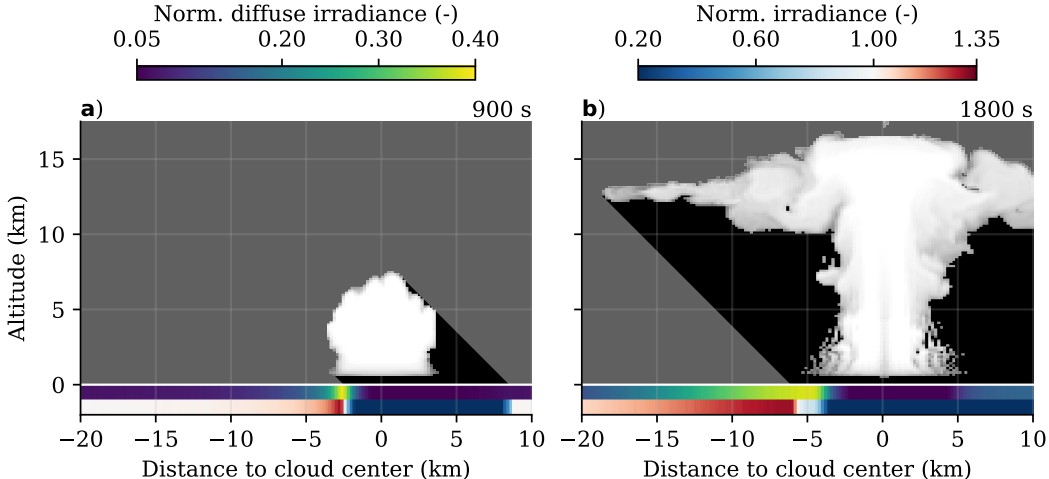

**Figure 14. SSI peak location and the effect of anvil shading.** An example surface irradiance pattern from the shear-free cumulonimbus simulation is shown at two time steps, one with and one without anvil shading. Solar zenith angle is $45°$, surface albedo is 0. The cloud is visualised by a liquid and ice-water cross-section, taken exactly halfway through the cloud in the y-direction. Dark grey shading illustrates the area where direct irradiance is 0. Normalised with clear-sky surface solar irradiance.

Tilting the updraft (with vertical wind shear) does not qualitatively change the SSI pattern with respect to the shear-free simulation, as illustrated in Figure 15. However, the relationship of domain averaged diffuse enhancement with cloud depth is not linear anymore during the updraft growth. Likely, the tilt results in a non-linear relationship between total sunlit cloud area as function of cloud depth, unlike the shear-free and idealised versions of this experiment.

Solar azimuth angle becomes important in this simulation, since the simulated cumulonimbus is not rotationally symmetric anymore, especially once the anvil develops and gets advected downwind, away from the updraft base. With the Sun in the west (upwind), there is no interference from the anvil cloud, and thus the peak irradiance enhancement underneath the cloud edge on the sunlit side remains. Illumination on the eastern (downwind) side of the cloud is quickly reduced by anvil shading,

but irradiance enhancement remains large as the anvil spreads out. With the Sun in the south, the updraft base stays sunlit with a slight extension of the anvil, and so total irradiance enhancement on the sunlit side is among the highest. There is no clear relationship between solar azimuth angle and total diffuse enhancement, however. We think any differences will relate to the specific shape of the cloud and its orientation with respect to the solar angle, and hence will not generalise beyond this case.

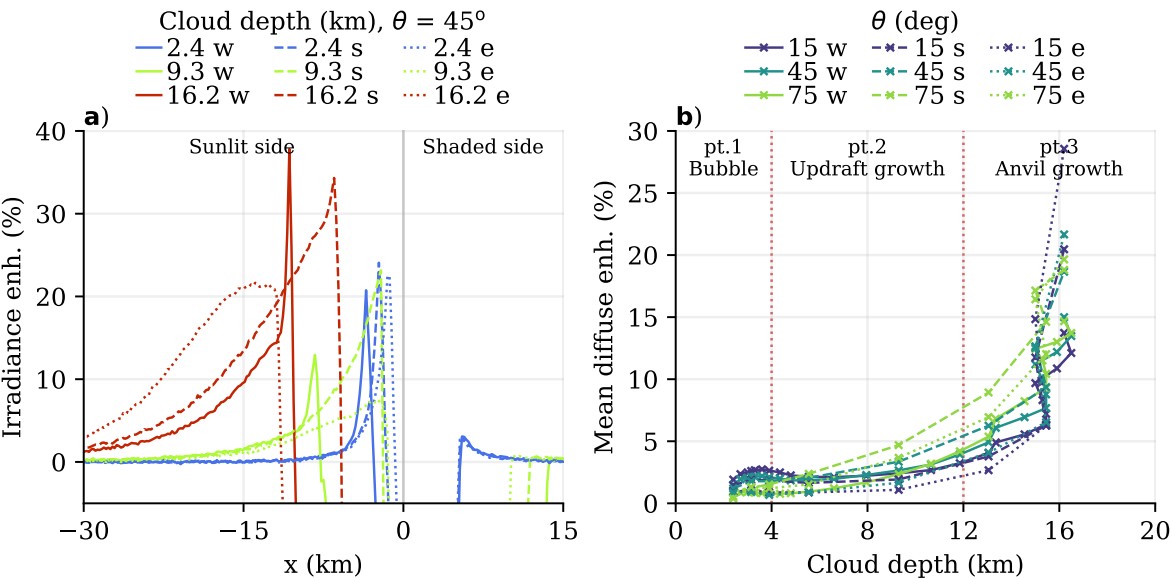

**Figure 15. The effect of wind shear and solar azimuth angle on SSI in a simulated cumulonimbus.** The surface diffuse enhancement is shown in **(a)**, similar to Figure 13. Now, the x-coordinate is centred in the middle of the domain rather than the cloud centre or side, so the pattern shifts as the cloud moves or dynamically evolves, and as the solar azimuth angle changes. The Sun shines from the west, south, or east, labelled with w, s, and e, respectively. Data is rotated such that the x-coordinate follows the solar azimuth angle.

## 5.6 Free convection integrates all mechanisms

All clouds presented so far are either flat and horizontally structured, or vertically structured but isolated. In the last experiment, we want to give a first look at what happens under highly complex cloud fields, for which we use the simulated `free convection` case. After three hours of simulation time, the cloud field features a handful of deep convective clouds with large anvils, about as many newly developing cumulus congestus, and numerous cloud remnants of previous convection distributed throughout the domain. This cloud field and resulting SSI field for an intermediate solar zenith angle of 45° is shown in Figure 16.

One similarity with the cases of isolated deep convection is that peak irradiance enhancement occurs on the sunlit side of deep convective clouds, but in this case only when there is no shading from other nearby clouds. In regions furthest away from clouds, the enhancement decreases to a local minimum of ~ 10 % (e.g., x = 25 km, y = 80 km). Regions with the highest

irradiance enhancement are enclosed by an optically thick updraft to the east and a high degree of cloudiness in the other directions (e.g., x = 20 km, y = 20 km). *Forward escape* plays a partial role in this, but it can originate from multiple clouds and cloud layers. To mark these parts of the cloud field, we show areas where $\tau < 6$ at an altitude above 7 km in blue, and in red when below 7 km in Figure 16b. These cover 35 % and 10 % of the domain, respectively, and can overlap. Other cloud areas have high enough optical thickness for *albedo enhancement* to contribute.

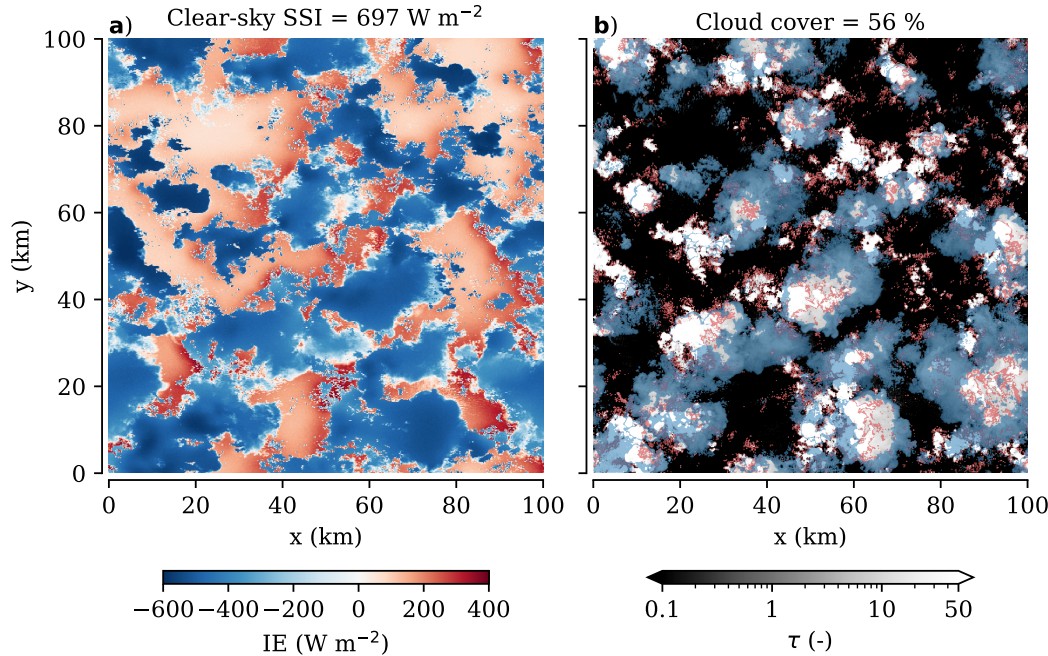

**Figure 16. Irradiance enhancement in free, deep convection.** Total surface solar irradiance relative to clear-sky is shown in (**a**), for a solar zenith angle of 45°. Vertically integrated cloud optical thickness $\tau$ is shown in (**b**). The time step is 10800 s, also shown in Figure 7. Clouds with $\tau < 6$ at an altitude above 7 km are marked in blue, or in red when below 7 km, to indicate regions where *forward escape* occurs.

What this simulation shows is that all four mechanisms can be active at once and be further complicated by having multiple deep convective clouds. The latter can be destructive to peak irradiance enhancement by anvil shading, or shading by other clouds, but also constructive by providing additional diffuse irradiance from multiple cloud edges towards areas already strongly enhanced by a sunlit side of a cloud base. Finding an analogue in observations is difficult, with the cloud field being this complex and the SSI patterns this large.

## 530  6   Conclusions

In this study, we performed numerical experiments to understand which main mechanisms drive surface solar irradiance extremes across a diverse set of prototype cases based on observations. We formulated four mechanisms, based on observations

and prior research, that can explain the observed irradiance extremes: *forward escape*, *downward escape*, *side escape*, and *albedo enhancement*. First, we will synthesise the results of all the experiments. Then, we will discuss some limitations and potential future research directions, with a first look at what happens in the complex, multi-layered cloud field of unorganised deep convection.

## 6.1 Synthesis of results

For clouds or cloud fields that are much wider than tall, such as stratus or altocumulus, we find that the mechanisms driving SSI variability depend on cloud optical thickness $\tau$. The transition from dominantly *forward escape* to *downward escape* is estimated to be occur between $\tau = 5.4$ and $6.3$, for high and low surface albedo, respectively. In the presence of optically thin clouds ($\tau < 6$), *forward escape* drives enhancement of diffuse irradiance, which largely follows the path of direct irradiance. Cloud altitude determines the extent of horizontal smoothing of the forward scattered irradiance. It leads to smaller areas of higher extremes at lower altitude compared to more uniform diffuse irradiance from scattering at higher altitude. Increasing the cloud area increases the total amount of scattering, which explains the extremes observed and simulated in cloud fields with gaps like altocumulus.

For optically thick clouds ($\tau > 6$), irradiance is predominantly scattered diffusely downward, leading to irradiance extremes near cloud edges or cloud gaps. In the transition zone ($\tau \sim 6$), both *forward escape* and *downward escape* contribute to the irradiance extremes.

Once *downward escape* starts to play a role in SSI variability, the clouds are optically thick enough to produce multiple scattering events between the surface and cloud. Under high albedo conditions and optically thick cloud cover, diffuse irradiance is then further enhanced by surface albedo, and accounts for 10 to 60 % of the total irradiance enhancement depending on low (0.2) or high (0.8) albedo conditions, respectively. The `checkerboard` case shows little contribution of *downward escape*, thus *albedo enhancement* is negligible and only contribute to a further 10 % increase in diffuse irradiance for the highest surface albedo conditions (0.8).

For isolated deep convective clouds, the sides of the cloud act as a region where photons diffusely escape. Areas $\sim 20$ km away of the sunlit side of the cloud can still have weak irradiance enhancements ($\sim 5$ %) prior to anvil formation. Underneath the edge of the cloud base, at non-zero solar zenith angles, the peak enhancement can reach up to $\sim 35$ %, and is driven by *downward escape* and *forward escape* rather than *side escape*. The anvil shading the updraft of the cloud removes the local irradiance extreme found near the sunlit cloud edge and instead greatly amplifies the area of irradiance enhancement.

## 6.2 Outlook

We have purposefully focused the majority of this study on understanding observation-based prototype cases of SSI variability that occur in single-layer horizontal or isolated vertical cloud fields. One reason for this is that observing the 3D structure of more complex cloud fields is very difficult, as ground-based observation typically can not see beyond the bottom of the first layer, and satellite observations not beyond the top of the highest layer. Reconstructing a prototype case to then numerically simulate brings too much uncertainty with it to disentangle the extra complexity that such a 3D cloud field provides. This is

another way of saying that we do not have the observations to validate the case of simulated `free convection`, contrary to all other simulations in this study.

However, a next step would indeed be to expand this research to multi-layered cloud fields and non-isolated deep convective clouds. Multi-layer cloud fraction is estimated to be between 10 and 50 % globally (Li et al., 2015), highest around the equator and mid-latitudes, and mostly because of high clouds. It would be interesting to find out whether multi-layered cloud fields are additive in their effect on SSI, or if entrapment between layers results in significant non-linearity. As for non-isolated deep convection, a first analysis we did into this direction suggests such cloud fields produces irradiance extremes from a combination of all mechanisms formulated in this study. Furthermore, there is the additional effect of multiple clouds constructively or destructively combining their effects on SSI.

To better quantify how much each mechanism contributes to the total SSI variability, it would help to keep statistics of scattering events and direction of travel of photons. This is possible with ray tracing, but simply is not implemented in our model. Instead, we estimated the transition between *forward escape* and *downward escape* by sampling the diffuse SSI field given a cloud. With the extra statistics, however, this transition between mechanisms can be more precisely defined.

We have not separated the effects of liquid and ice phase condensate in our analyses, but the contribution of each scattering mechanism may depend on cloud droplet phase as well. For example, this may be relevant for the cloud tops and anvils of cumulonimbus clouds in the upper troposphere, or in mixed-phase clouds, as is often the case in altocumulus (Barrett et al., 2017). The simulations with deep convection do not suggest there is a significant change in how the mechanisms behave once ice appears, however.

From all experiments, we conclude it is primarily the overall 3D cloud geometry and optical thickness that determine how the SSI is affected. Sensitivity experiments where we change the scattering phase function or number concentrations of water and ice particles support this conclusion, as patterns qualitatively remain unchanged (see Appendix).

In summary, the findings of our experiments provide a foundation to understand which mechanisms are at play for any given cloud or cloud field that is either flat and horizontally structured or isolated and vertically structured. We believe this is a good starting point for future analyses of surface solar irradiance under cloudy conditions, whether those are simple but more observationally constrained conditions than our prototypes, or perhaps maximally complex like unorganized deep convection. In any case, the diversity of SSI patterns and extremes with the set of cloud fields we have demonstrated should motivate others to consider cloud fields beyond a single type and perhaps also include the effects of surface albedo.

*Code and data availability.* BSRN time series data is described in Mol et al. (2023b), and is available in Knap and Mol (2022) and Mol et al. (2022). Spatial SSI data of FESSTVaL is described in Mol et al. (2024), and is available in Mol et al. (2023a). Model code, setup files, output data, Veenkampen time series data, spatial SSI patterns, and python scripts to analyse and visualise the data are available at https://doi.org/10.5281/zenodo.14652241.

*Author contributions.* WM designed the methodology, performed the analyses, wrote the manuscript, and curated the data. CvH conceptualised the overarching research aim, acquired the funding, reviewed the methodology and writing of the manuscript, and provided the resources to run the experiments.

*Competing interests.* The authors have no competing interest.

*Acknowledgements.* We thank Mirjam Tijhuis, Menno Veerman, and Bart van Stratum for their input in discussions on radiative transfer and for technical discussions on how to set up MicroHH and the ray tracer in the best way for this study. Also, we thank two anonymous referees for their useful and constructive feedback, and lastly, Georgi Yordanov for his feedback on our experimental design and interpretation regarding the cloud gap experiments. The authors acknowledge funding from the Dutch Research Council (NWO) (grant: VI.Vidi.192.068).

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

# Appendix A:  Sensitivity analyses

## A1    Phase function: Mie vs. Henyey-Greenstein

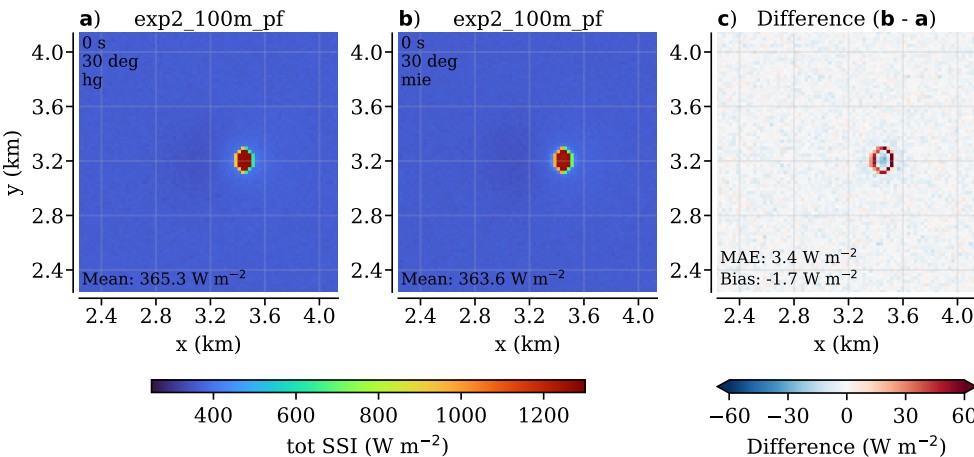

**Figure A1. The effect of phase function choice** in the `cloud gap` case with solar zenith angle set to 30°. Subplot **(a)** shows the total SSI using Henyey-Greenstein (HG) and **(b)** using Mie look-up tables. The difference in **(c)** shows the increase in SSI at the project cloud gap edge and slight decrease just around it, highlighting that Mie resolves the narrow forward scattering peak whereas HG is more diffuse.

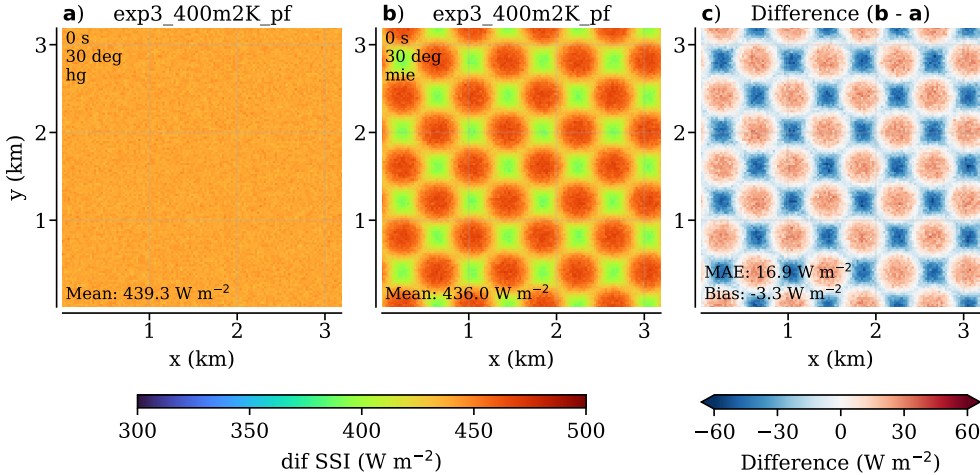

**Figure A2. The effect of phase function choice** in the `checkerboard` case with solar zenith set to 30°. Same layout as Figure A1, but for diffuse SSI.

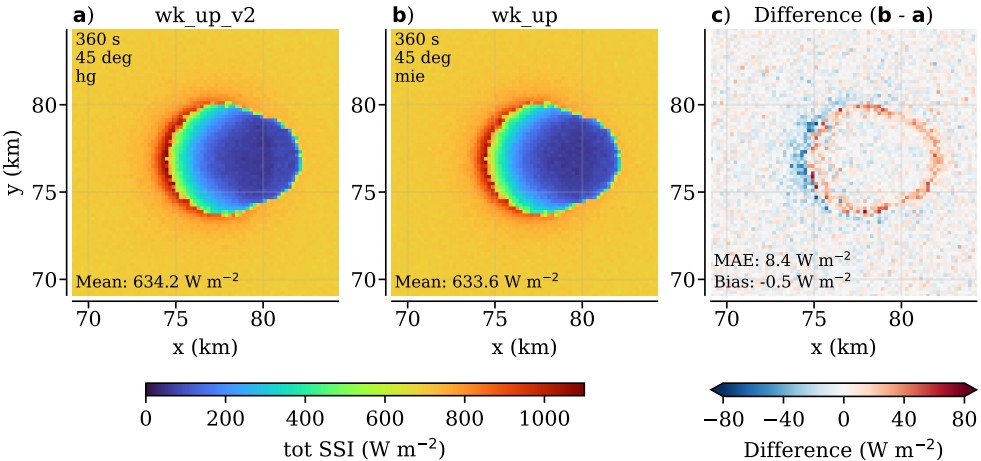

**Figure A3. The effect of phase function choice** in the shear-free cumulonimbus case with solar zenith set to $45°$. Same layout as Figure A1.

## A2 Droplet number concentration

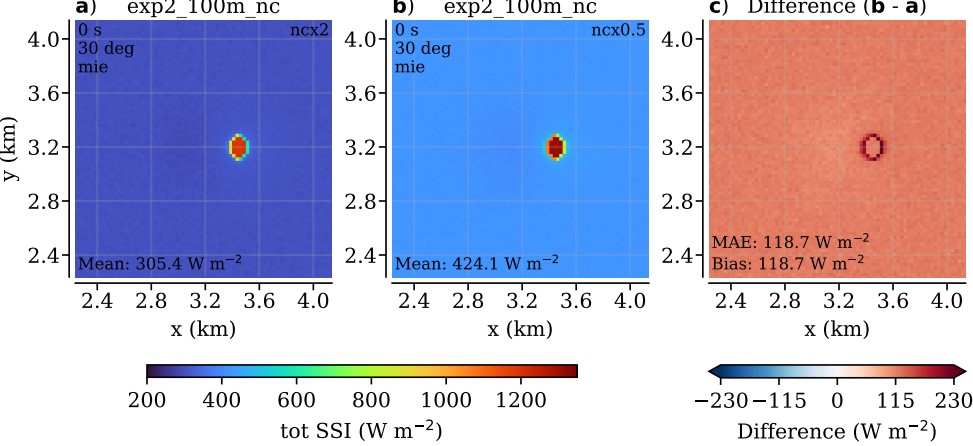

**Figure A4. The effect of changes in droplet number concentration (nc)** in the `cloud gap` case with solar zenith set to $30°$. All simulations use the Mie phase function. The nc is doubled (`ncx2`) and halved (`nc0.5`) relative to the standard simulations (200e6 and 50e6) and shown in respectively **(a)** and **(b)**. The rest of the layout is the same as Figure A1. The increased effective radii in **(b)** compared to **(a)** result in a lower optical thickness, brightening the whole scene shown in **(c)**.

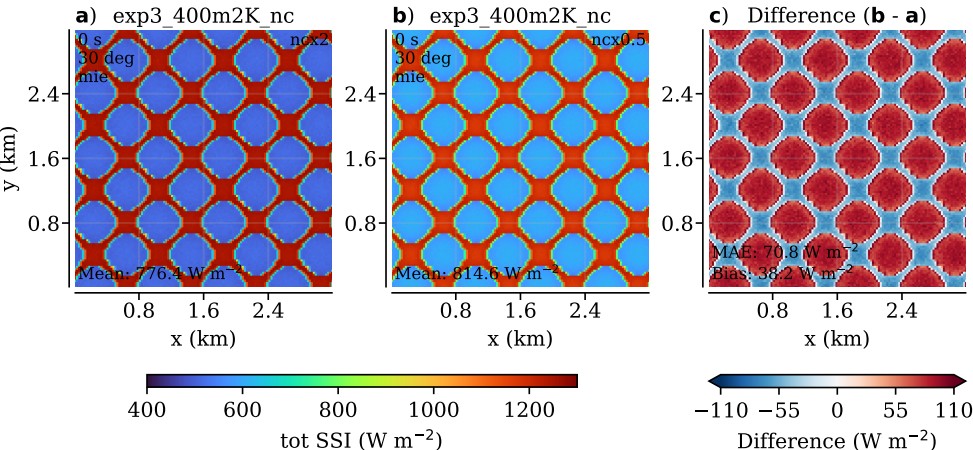

**Figure A5. The effect of changes in droplet number concentration (nc)** in the `checkerboard` case with solar zenith set to 30°. Same experiment and plot layout as Figure A5. The increased effective radii in **(b)** compared to **(a)** result in a lower optical thickness and less diffusive forward scattering, brightening the projected shadows and darkening the sunlit areas, shown in **(c)**.

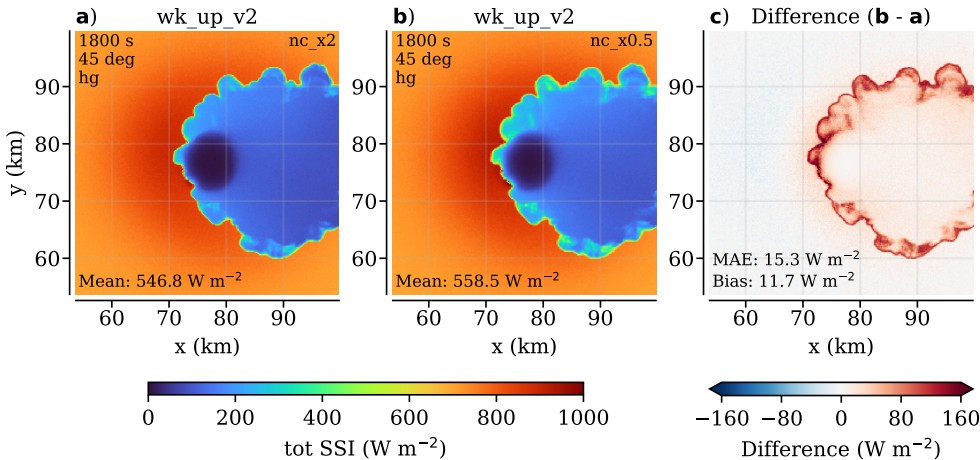

**Figure A6. The effect of changes in droplet number concentration (nc)** in the shear-free cumulonimbus case with solar zenith set to 45°. Here, we use the Henyey-Greenstein phase function. The nc for both ice and water are doubled and halved relative to the standard simulations (200e6 and 50e6 for water, 2e5 and 5e4 for ice). Same layout as Figure A5. The increased effective radii in **(b)** compared to **(a)** result in a lower optical thickness and less diffusive forward scattering, brightening the projected shadows and darkening the sunlit areas, shown in **(c)**.