# Peer review of "Mechanisms of surface solar irradiance variability under broken clouds"

_EGUsphere, 2024_

## Referee Comment (RC2)

Review of the paper "Mechanisms of surface solar irradiance variability under broken clouds"
by Wouter Mol and Chiel van Heerwaarden, DOI: 10.5194/egusphere-2024-2396

This paper focuses on surface solar irradiance (SSI) variability in broken cloud situations and investigates mechanism for SSI enhancements in presence of broken clouds. Therefore, 3D radiative transfer simulations for idealized and non-idealized clouds were performed and resulting irradiance fields are systematically evaluated to explore the involved mechanisms.
This paper is generally well written, presents new insights and can be a valuable basis for further studies in this field. I believe it is suited for publication after addressing my comments below and major changes to the introduction.

General comments

L. 15-30: The introduction is currently not matching the overall quality of this paper and seems to be suitable only if placed within a larger academic work ("thesis", L.17). For this article, I would suggest a general introduction to the overall topic including its importance, placing this study in the context of prior work and describing the scope and additional value of this study instead of starting with a description of the content of the introduction chapter.
While chapter 1.2 gives some references to previous work, it also introduces a concept for separating relevant mechanisms and gives fundamental definitions for this work. I would therefore suggest to move this to a separate chapter outside the introduction.

Overall, the language and figure descriptions seem unprecise in multiple occasions. Some important information and details for reproducibility are missing. Although the provided source code and data could give some hints, I think the quality of the paper would profit from some thorough revision. In the specific comments below, I give examples, where in my opinion improvements could be easily adapted.

I encountered problems opening the "model_data.zip" provided through your zenodo data publication on a linux computer. Please double-check the file is usable.

Irradiances obtained with MCRT are subject to uncertainty. While this uncertainty might not be crucial for the results presented in this study, at least an order of magnitude should be mentioned in appropriate places of this paper. This could at least be a general upper limit desired for all experiments or experiment specific. The uncertainty/noise is well visible in ,e.g., Figs. 8, 9 and following.

Specific comments

L. 3:        "surface solar irradiance extremes": This work and the mechanisms are about *maxima*, minima are also extremal but not discussed. While I do not see this as critical in this occasion, please think about more precise wording in general. This would apply also the the title, as "variability" includes a lot more than the irradiance enhancement mainly discussed in this paper.

L. 4:        Missing word after "and" (low?)

L. 29:      Missing mention and description of Section 4 and 5

Fig. 1:      Missing legend for color coding of lines

L. 48:      Is there any previous work or citable resource, on increased cloud cover fraction of altocumulus compared to shallow cumulus? If yes, please include a reference else I do comply with this feeling and see that this is more of a definition for this work, but you may think about a less general formulation.

L. 75:      "clear-sky to overcast conditions" seems misinterpretable to me, potentially including all conditions in between. I would suggest 'the transition from clear-sky to overcast conditions' instead

L.76:      "Example[s]"

Fig. 2:      Colorbar label is "Normalized" while "normalised" is used in the caption and mostly throughout the document.

Fig. 2:      The axes do not really match what the figure describes, as the independent patches do not have an obvious spatial relation on x- and y-axis. I would suggest just scales in "cross-wind" and "along-wind" direction for more clarity.

L. 88:      Probable typo: "I[n] all cases"

L. 94:      The "region" of optical thickness seems misinterpretable to me as a spatial description, as area and value range are important throughout this study. Possibly, using 'range' instead could clarify this a bit more.

L. 102:      "optically thin area": "area" (L. 97) refers to surface area and "parts of" (L. 100) as well as "sections" (L. 101) refer to spatial distribution of optical thickness. More stringent nomenclature could benefit readability a little bit here.

L. 105:      What is the "upper limit of optical thickness" referring to here?

L. 107:      "creat[ing]"

L. 150:      As longwave is not used for this study to my understanding, the mention of the "128 set for longwave" seems irrelevant. I suggest either not mentioning it, in case you did not use and potentially also did not compute it or explaining why you needed to compute it, as neglecting it would have saved significant amount of computational effort.

L. 181–184: The text should mention that the cloud is only populating half of the domain. Information like domain size and horizontal resolution is also missing in the text. Also information on solar azimuth would clarify the setup. There is no information on atmospheric properties apart from clouds. For clear-sky SSI values (which then could be mentioned for example here) and reproducibility, this is a necessary information. Also the (periodic?) boundary conditions of the MCRT are only mentioned later on for the Cb-case.

L. 185:      Fig. 4 suggests the domain is 12.8km x 12.8km for stratus, but 6.4km x 6.4km for cloud gap. Is this "the same configuration" and only an excerpt shown or do the domain sizes differ? Please

clarify.

L. 189: Two times "manually": I suggest deleting the later occurrence

L. 196-201: As later use of this case suggests, the domain size here is not 12.8km x 12.8km (or 6.4km x 6.4km), but larger to isolate the towering cumulus. Please mention the actual domain size also here. While it is later on (Section 4.5) noted, that the optical thickness is ensured to be large enough, a thorough documentation of your scaling of optical thickness with increasing cloud depth is missing and should be added, for example here.

L. 208-209: Repetitive use of "altocumulus", e.g., the second occurrence could be replaced by 'This'.

Fig. 6: As cloud depth is later on (e.g. Fig. 13) used to distinguish cases, it would be nice if this would be given for the displayed snapshots for a better association.

L. 244: I am unsure whether there should be a second "in" in "for the scattering regime we are [in] in terms of mechanism".

Fig. 9: The domain cross-section used for this plot could be indicated in Fig. 4 to get a feeling of the averaged region in y-direction, especially in the checkerboard case. This would also give a hint on sun direction there. Otherwise "(part of)" (Fig. 9, caption L. 2) is a very vague definition.

Fig. 9: While theoretically reconstructable from Norm. SSI (-), AE (W/m²) and AE as % of IE for the reader, it would be helpful for the reader to have clear-sky SSI (dir/dif/tot) values given at least for this figure or more in general in the text.

Fig. 9 g-i: The extremal values on the shadow borders are a striking feature and should at least briefly assessed in the text. Also, there is no explanation of the grey regions here.

Fig. 9 i: In the case of sun zenith angle SZA=30° the gray shaded area including the border grid boxes do have an x-extent of about 2x the extent of the valid data area excluding the border grid boxes with extremal values. To my understanding, of the text, the cloud disks are 500m in diameter with a maximum distance of 150m in between (L. 193-195). This would suggest a ratio of > 3:1. I guess this is due to the selected and averaged y-axis range, but this needs more explanation.

L. 304-310: A full description of the setup would in my opinion include albedo and SZA in the text, not just in Fig. 10

Fig. 10 and Section 4.4.1:
      Is the SSI summed/averaged over the entire domain (almost) constant for all cloud altitudes and therefore the power only redistributed or is there a significant difference in surface solar power based on cloud altitude? To me, this would be a nice additional information and support the previous explanation of checkerboard case SSI.

L. 326: Can omit the "in" in "very small [in] for the simulated altocumulus"

L. 336: "There are two exceptions where forward escape still occurs": In a statistical sense, individual photons/rays can always be scattered only once or twice and therefore there is always some fraction of forward escape. While the meaning is fully understandable in this context, please

consider rephrasing to avoid the impression of exclusivity of the mechanisms. I would suggest replacing "occurs" by 'plays a significant role' or 'contributes significantly'.

Fig. 13 caption:
Neither a nor b do directly show SSI patterns as indicated by the caption. Please make the caption, especially the first sentence, more precise.

Fig. 13 a:     Are 17.2 and 18.4 ordered on purpose like this in the legend?

L. 371:     There is still a maximum in IE in these cases, it is just spatially shifted and/or decreased. Please rephrase "disappearance" or specify the "peak irradiance enhancement" you mean more in detail.

L. 377:     For example 'hinder' or 'obstruct' would describe the process more factual than "takes over the side escape mechanism". At the moment this sentence does not reflect the actual process to me.

Fig. 14:     Cb cases were before referenced by LES time or cloud depth. For better orientation, it would be helpful to get this information here as well.

Fig 15 caption, L.3:
"The [S]un"

L. 399-400: The regions meant here are the ones away from 'cloud shadows' and not from "clouds", I suggest? Please consider rephrasing.

L. 409:     "sunlit cloud base" sounds to me only possible with SZA > 90°. Please clarify.

L. 427:     Also for clouds with optical depth > 6, a (small) fraction of light is scattered only once or twice. Therefore I suggest adding "irradiance is [predominantly] scattered uniformly downward".

L. 427:     To my knowledge, the downward radiance distribution underneath optically thick clouds is not "uniform". Please clarify.
This can be looked up, e.g., in theory and simulation in Sobolev (2017),  Grant et al. (1995), in measurements in Nagata et al. (1997) or simply using a 1D-RT calculation with an optically thick cloud.

L. 439:     "onto" seems the wrong word here to me. Perhaps use 'of' instead?

L. 460:     Can omit "out".

L. 469:     Following the sentence structure, I believe it should be 'maximally complex' instead of "maximum complexity".

References

Sobolev, V. V. (2017). *Light scattering in planetary atmospheres: international series of monographs in natural philosophy* (Vol. 76). Elsevier.

Grant, R. H., Heisler, G. M., & Gao, W. (1996). Photosynthetically-active radiation: sky radiance distributions under clear and overcast conditions. *Agricultural and Forest Meteorology*, *82*(1-4), 267-292.

Nagata, T. (1997). Radiance distribution on stable overcast skies. *Journal of Light & Visual Environment*, *21*(1).

---

## Author Comment (AC1)

Dear Johannes Quaas and both referees,

First, thank you to both referees for your (many) helpful comments. We will post our response to your comments as individual replies and address major changes here first. All our text is in blue.

It has taken relatively long to reply and write a revision due to the discovery of several bugs relating to the treatment of ice in our simulations. This required careful re-running of simulations and re-interpretation of some results.

In this Author Comment we first summarize four major changes since the initial submission:

1. Changes due to bug fixes in the radiative transfer calculations
2. Introduction of sensitivity experiments
3. Rewriting of the introduction sections
4. Revised interpretation of 'cloud gap' experiments

On behalf of both authors,

Wouter Mol

**1. Bugs in radiative transfer calculations involving ice**

There were 3 bugs in the radiation code that noticeably affected all simulation with ice particles (figures 13, 14, 15, and 16):

1.  The effective radius for ice was cut off to the limits for liquid water due to a bug in the code, and therefore optical properties and phase function were biased towards smaller droplets
2.  Mie phase functions for ice particles was not implemented, the code silently switched to the lowest value for water droplets. Scattering through ice was thus effectively too diffuse
3.  Ice optical property in the upstream RTE+RRTMGP lookup tables were defined in diameter, but were marked as radius in the metadata. . This resulted in effectively using too small effective radii, making ice clouds more optically thick

Bugs 1 and 3 have been fixed and bug 2 avoided by using Henyey-Greenstein (HG) in favour of untested Mie tables for ice.  Simulations containing ice were performed again, which means figures 13, 14, 15, and 16 are updated with new data.

The combined effect of all fixes is that ice clouds now have a visibly lower optical depth than before. This has brightened up the area under anvil clouds by 10% and further increased the sunlit side any updraft. See the figure below for an example.

[Figure]

*Figure 1: old (a) compared to new (b) before any ice formation in the deep convection simulation without wind shear. This only shows the effect of HG vs. Mie phase function as there are no ice clouds yet present.*

[Figure]

*Figure 2: old (a) compared to new (b) simulation illustrating how the anvil cloud (all ice) has become more optically thin.*

**2. Introduction of sensitivity experiments**

The simulation-related issues and questions by the referees (see below) prompted us to run a set of sensitivity experiments to test the robustness of the simulations in response to the choice of phase function (Mie vs. HG) and droplet number concentration (indirectly effective radii). New figures are added to the appendix of the revised manuscript and are discussed in the methods section 3 and results section 5.

The text describing droplet number concentrations has been changed to reflect actual values being used:

- Nc0 was hardcoded to 1e8 for water and did not use the value in the .ini as reported originally (2.5e8)
- Nc0 was different for water and ice, and therefore the effective radius was too. This is now clarified: Nc0 = 1e8 for water and 1e5 for ice by default.

**3. Revised introduction and background sections**

Our original introduction, Section 1.2, and Section 2 fell short in introducing and motivating our work in the right context, as pointed out by both referees. The introduction has been largely revised, the details of which we include in our response to the referees.

**4. Different interpretation of 'cloud gap' experiment**

We received comments from Dr Giorgi Yordanov, whose work we cite in relation to the cloud gap experiments. From this we learned that the way we design our cloud gap simulation limits the potential of forward escape to generate highly focused extreme enhancement (up to x1.8 clear-sky), due two factors: too high optical thickness (downward escape regime) and cloud gap radius well above the apparent size of a solar disk.

Clouds may be configured such that a gap has an apparent size similar to the solar disk (0.5 degrees), the optical thickness is optimal (tau = 3.1 is their theoretical optimum), and with unobstructed direct irradiance. This would theoretically create a ring of cloud edges that creates a focused area of enhanced irradiance due to forward scattering. The figure below indirectly illustrates this. Making the gap small enough to focus this ring into one point would also close the gap to direct irradiance, unless the simulation is run at meter-scale resolution and the Sun is just a few degrees away from the zenith. We deem such a configuration unrealistic, despite it being an interesting theoretical limit.

We now clarify the difference between our experiments and that of Georgi Yordanov. We believe that small gaps in stratus or between altocumuli result in SSI patterns with significant irradiance enhancement exactly as already stated in our initial submission. However, we add that under rare conditions we are underestimating SSI maxima due to resolution, which limits the potential for forward escape to add more SSI focussed on the sunlit area.

[Figure]

Figure 3: Phase function choice highlights a ring of enhanced SSI for Mie scattering relative to Henyey-Greenstein (this is new Figure A1)

---

## Author Comment (AC2)

In my opinion, this paper presents an interesting analysis and provides new insights that can help us better understand cloud-related enhancements in surface irradiance. The proposed theoretical classification makes sense, the methodology is suitable for the task, and the analysis is thorough. (I believe this even though, as mentioned near the end of the manuscript, the study does not use the most powerful analysis approach of examining photon paths and scattering directions.) The presentation is of a generally high quality, but important improvements are still needed, most critically in the introduction section. My specific comments are listed below.

Thank you for your constructive and helpful comments. We have made several larger changes as stated in our general reply and smaller ones as described here below, which hopefully properly addresses your concerns.

**Major issue:**

The introduction section needs a thorough revamping, for several reasons.

First, the introduction should provide context and historical perspective to the presented study. For example, it should address the following questions. Did other researchers previously examine (using observations and/or theoretical calculations) cloud-related surface irradiance enhancements, and what were their main findings about the frequency, magnitude, sources, and consequences of these enhancements? What is the underlying motivation for us to care about these enhancements: Is it perhaps something about solar energy production or the health risks of UV radiation, etc.? Did the earlier results leave major gaps that we still need to fill, perhaps in observing, understanding, simulating, or predicting the enhancements? Which of these gaps does the current paper help us fill?

These are all valid questions, some of which were not answered adequately in the initial submission while some others were not logically placed and were therefore easily missed (scattered through original Subsection 1.2 and Section 2). For clarity:

- The introduction section now covers the context of this research including our motivation for doing this study and (some of) the impacts of radiation variability
- We now more clearly state that we discuss the previous work (observations, modelling, theory) on 3D radiative transfer and mechanisms of variability in Section 2 (definition and examples of variability) and Section 3 (our proposed mechanisms).

- We discuss previous work more clearly and include a few more studies that were originally missing

Second, Section 1.2 does not seem introductory, as it proposes a new theoretical framework that is a key element of this study. Therefore, I recommend moving Section 1.2 into a new section of its own.

We have revised the structure of the introduction after rewriting most of it and moved original subsection 1.2 to a separate section as suggested.

Third, the introduction (just before the start of Section 1.1) presents a brief preview of what we can expect in each section of the paper, but this preview stops at Section 3 and does not include sections 4 and 5 (which present the results and conclusions, respectively).

This mistake has been fixed, thanks for noticing.

Finally, the very first sentence starts the paper off by telling about the introduction section rather than about the paper as a whole, and by referring to a "chapter" and a "thesis" (which suggests that the text was simply copied from an academic thesis).

Regarding 'chapter' and 'thesis': this manuscript was written as a final content chapter of my thesis, in parallel to my thesis introduction and discussion chapters. Evidently, I haven't managed to keep the references in this ACP version separate from the thesis. I'm not sure how we missed the mistake in the first sentence... In revising the introduction, I have taken care of the incorrect references.

**Minor issues:**

Line 3: The wording should make it clear that the paper covers only extreme highs and does not discuss extreme lows.

This is now changed to "variations" rather than "extremes". While extreme highs is a large part of our focus, "variations" more completely covers our analyses.

Lines 8-9: The wording should be changed because as is, it discusses an albedo effect but not discussed a mechanism. Perhaps trapping or multiple reflection between surface and clouds could help in phrasing the albedo-related process as a mechanism.

True. Upon re-reading the abstract, we found some other conclusions that were not summarized sharply either. We have rephrased most of the abstract while not changing the conclusions we draw therein.

Figure 1: The caption or a newly added legend should explain what the lines of various colors represent.

Indeed, there should have been a legend. This is now fixed.

Line 88: Just a typo: The correct start to the sentence should be "In all cases…"

Fixed.

Line 139: It would help to clarify the main between between Monte Carlo ray tracers and radiative transfer models.

I have dedicated a sentence to briefly clarify the main difference (Section 2).

Line 147: It should be clarified what RTE-RRTMGP stands for (I imagine RTE is for Radiative Transfer Equation) and, if possible, a reference should be given.

I have added a reference to where this model is originally described, but would rather not give the meaning of the long acronym. It is a name of a model and otherwise adds little information to the reader that is relevant to this study: "Radiative Transfer of Energetics – Rapid Radiative Transfer Model for General circulation models – Parallel". There are now two references to cover the origin of the model: Veerman et al. for the Monte Carlo implementation and Pincus et al. for the RTE+RRTMGP reference, and I describe the model as "radiative transfer solver".

Line 150: Are longwave simulations used in this study? If yes, it should be mentioned what they are used for; if not, they should not be mentioned.

They are used in the online 1D radiative transfer calculations for the simulated altocumulus case, but not used in any analyses. I have removed the mention of longwave here.

Line 158: What are the wavelength limits of the used visible spectral band?

625 to 768 nm. I could use other bands in more energetic parts of the spectrum (more green and blue wavelengths), but the difference in tau is small (< 5%) and much of the energy is in longer wavelengths too. The text was ambiguous, but 'most energetic' was

meant to refer to the visible spectrum as a whole, not the specific band chosen. The text is now clarified, and it also includes the wavelength limits and a bit of context for how much optical thickness varies.

Figure 8: In the caption, it would help to clarify what exactly is meant by "relative to clear-sky values"; I guess it's clear-sky values of diffuse irradiance rather than clear-sky total irradiance. Also, it could help to explain why, in the right-side plot, the area under the cloud is white. (Alternatively, could the location of the cloud be marked by a circle and allow us to see the enhancement inside the circle?) Finally, it might help to clarify in the caption or around Line 245 that the yellow dashed line is not visible in the left side plot simply because it coincides with the solid line.

These are all good suggestions, thank you. 'Diffuse' added to 'clear-sky', changed the white circle to a line, described what the line means in the caption, and added that albedo has no effect at low tau in the text. I also added (a) and (b) subplot labels.

Line 270: The word "under" could be changed to something like "in cases of", as most values under clouds (i.e., shaded areas) are blocked out by grey in the key, lowest row in Figure 9.

I disagree. Figure 9d,e,f show clearly how irradiance is strongly enhanced specifically only under optically thin clouds.

Figure 11: It should be explained what the "diffuse peak probability" (shown in dashed lines in the right-side column) is.

It is the most probable value of diffuse irradiance. This was awkwardly phrased. I now say "most probable diffuse irradiance". Adding the PDF to these plots would make it much harder to read, unfortunately.

Figure 12b: It should be clarified whether optical depth increases when cloud depth increases, or the optical depth remains unchanged, and the cloud gets less dense as it gets deeper. This could be clarified either around here or around Line 170.

This is now clarified in the case description in section 3.1.2. We don't rescale the liquid water over each deepened cloud, but rather copy the liquid water values upward. Optical depth and liquid water path therefore increase linearly with height, because all else is kept constant.

Line 354: The wording "scattered direct irradiance" seem self-contradicting, as direct irradiance is, by definition, non-scattered.

It refers to all the direct irradiance that is scattered rather than transmitted or absorbed. I do not find it contradicting, in the same way that 'condensed water vapour' is not self-contradicting: it

Figures 12 and 13 (and perhaps others) should be placed after they are described in the text.

Figures are now placed after they are referenced in text. One exception is Figure 5 which is referenced immediately after placement, hopefully it will fit better during eventual typesetting.

Line 370: It might be worth adding "and its immediate surroundings" after "itself", given the finding that, for 16.5 km cloud depth, some very high values occur outside the updraft.

I am not quite sure I follow. High peak IE values disappear once the anvil grows large enough to shade the updraft. At 16.5 km cloud depth in initial Figure 13a, this simply did not yet happen. After our ice-related bug fixes the difference in SSI between shaded and unshaded updraft is now less dramatic, however.

Lines 396-397: It seems worth mentioning that parts of the scene are shown in the right-side plot of Figure 7.

Indeed. I have changed the caption of Figure 16 to add the time step and therefore the cloud scene matches what is seen in the last time step of Figure 7.

Lines 420-421: I suggest either deleting the word "zone" or replacing the word "between" by "around".

I used 'zone' because it is not a hard threshold, but I agree this reads a bit strange. I have rephrased the sentence to "transition from dominantly forward escape to downward escape is estimated to occur between"

---

## Author Comment (AC3)

**Philip Gregor (referee #2)**

This paper focuses on surface solar irradiance (SSI) variability in broken cloud situations and investigates mechanism for SSI enhancements in presence of broken clouds. Therefore, 3D radiative transfer simulations for idealized and non-idealized clouds were performed and resulting irradiance fields are systematically evaluated to explore the involved mechanisms.

This paper is generally well written, presents new insights and can be a valuable basis for further studies in this field. I believe it is suited for publication after addressing my comments below and major changes to the introduction.

Thank you for your helpful and constructive comments, of which you had many. Some general themes were already addressed in an earlier part of this response, but please find our replies to each of your comments below.

**General comments**

L. 15-30: The introduction is currently not matching the overall quality of this paper and seems to be suitable only if placed within a larger academic work ("thesis", L.17). For this article, I would suggest a general introduction to the overall topic including its importance, placing this study in the context of prior work and describing the scope and additional value of this study instead of starting with a description of the content of the introduction chapter.

Agreed, the original introduction was much too concise and not introductory. We have thoroughly revised the introduction and modified the first sections to yours and the other reviewer's comments. I hope you will now find it better introduces the topic, motivation, knowledge gaps, and overall structure of the study.

As for the word "thesis", this was an awkward oversight, and indeed originated from writing my thesis introduction and outlook chapters in parallel to this study.

While chapter 1.2 gives some references to previous work, it also introduces a concept for separating relevant mechanisms and gives fundamental definitions for this work. I would therefore suggest to move this to a separate chapter outside the introduction.

Agreed, the lines between topic introduction, existing knowledge, and new knowledge were blurry at times. We have paid attention to this whilst revising the first sections of this manuscript. Section 1.2 has been moved to a separate section.

Overall, the language and figure descriptions seem unprecise in multiple occasions. Some important information and details for reproducibility are missing. Although the provided source code and data could give some hints, I think the quality of the paper would profit from some thorough revision. In the specific comments below, I give examples, where in my opinion improvements could be easily adapted.

Thank you for pointing this out. I think I have leaned too much on model description references and open data regarding reproducibility. The details are in replies to your specific comments, but overall, the experimental setup, model description, various figure captions, and textual comments to experiment details in the results section have been revised and are now more complete.

I encountered problems opening the "model_data.zip" provided through your zenodo data publication on a linux computer. Please double-check the file is usable.

I have the same issue. The .zip is apparently a .tar ("tar -xvf model_data.zip" gives you a collection of .zip files with each experiment), but it contains at least one corrupted .zip. I am not sure what happened, as my local copy is fine. I have updated the open data version with new model data and updated processing scripts, and I have verified that it works.

Irradiances obtained with MCRT are subject to uncertainty. While this uncertainty might not be crucial for the results presented in this study, at least an order of magnitude should be mentioned in appropriate places of this paper. This could at least be a general upper limit desired for all experiments or experiment specific. The uncertainty/noise is well visible in ,e.g., Figs. 8, 9 and following.

I think that presenting uncertainties of the MCRT based on statistical noise will not add relevant information. We have run all simulations to the point where the mean signal dominates the noise, as is visible in all figures. Of course, there are other sources of uncertainty, which we now try to demonstrate more transparently in a set of sensitivity experiments (phase function, droplet number concentration) and a better description of the methodology.

**Specific comments**

L. 3:     "surface solar irradiance extremes": This work and the mechanisms are about maxima, minima are also extremal but not discussed. While I do not see this as critical in this occasion, please think about more precise wording in general. This would apply also the the title, as "variability" includes a lot more than the irradiance enhancement mainly discussed in this paper.

Agreed on that the wording is sometimes used incorrectly. 'extreme' fits when discussing 'SSI variability', but not when referring to 'SSI' alone, as you point out. I have changed 'extremes' to 'variations' in L. 3.

Regarding 'variability', it is the best fitting word I have found to describe what I believe it is we are researching here. In original introduction (section 1.1), I define what I mean with 'variability', and while IE is a key part of this definition it not the only thing we discuss.

L. 4:     Missing word after "and" (low?)

"And" should not be there, this is now fixed.

L. 29:   Missing mention and description of Section 4 and 5

Fixed.

Fig. 1:  Missing legend for color coding of lines

Fixed and clarified figure caption.

L. 48:   Is there any previous work or citable resource, on increased cloud cover fraction of altocumulus compared to shallow cumulus? If yes, please include a reference else I do comply with this feeling and see that this is more of a definition for this work, but you may think about a less general formulation.

It was more an observation of our own, but we introduced it without any reference, nor is it a new insight. We have reformulated this and included a reference to a more general description of altocumulus.

L. 75:   "clear-sky to overcast conditions" seems misinterpretable to me, potentially including all conditions in between. I would suggest 'the transition from clear-sky to overcast conditions' instead

Agreed, fixed.

L.76:    "Example[s]"

Fixed.

Fig. 2:  Colorbar label is "Normalized" while "normalised" is used in the caption and mostly throughout the document.

Fixed.

Fig. 2: The axes do not really match what the figure describes, as the independent patches do not have an obvious spatial relation on x- and y-axis. I would suggest just scales in "cross-wind" and "along-wind" direction for more clarity.

I am not sure if I agree that they have no obvious spatial relation on an x- and y-axis if the directions are defined as along-wind and cross-wind. But the bigger challenge was to concisely show a collection of spatial patterns that are, indeed, independent. In the end I think I agree on the relabeling suggestion, but I've also moved the axes spines outward which to me gives a slightly better impression regarding the independence of patches.

L. 88:  Probable typo: "I[n] all cases"

Yes, fixed.

L. 94:  The "region" of optical thickness seems misinterpretable to me as a spatial description, as area and value range are important throughout this study. Possibly, using 'range' instead could clarify this a bit more.

Agreed, fixed.

L. 102: "optically thin area": "area" (L. 97) refers to surface area and "parts of" (L. 100) as well as "sections" (L. 101) refer to spatial distribution of optical thickness. More stringent nomenclature could benefit readability a little bit here.

I have rephrased multiple parts of the paragraph to clearly separate (Earth's) surface "area" underneath clouds and optically thin cloud "area". I expect this has improved readability.

L. 105:  What is the "upper limit of optical thickness" referring to here?

Optically thick or 'opaque' clouds, but this was phrased poorly. I have reformulated the sentence.

L. 107:  "creat[ing]"

Fixed.

L. 150: As longwave is not used for this study to my understanding, the mention of the "128 set for longwave" seems irrelevant. I suggest either not mentioning it, in case you did not use and potentially also did not compute it or explaining why you needed to compute it, as neglecting it would have saved significant amount of computational effort.

I have taken out the longwave g-point comment.

L. 181-184: The text should mention that the cloud is only populating half of the domain. Information like domain size and horizontal resolution is also missing in the text. Also information on solar azimuth would clarify the setup. There is no information on atmospheric properties apart from clouds. For clear-sky SSI values (which then could be mentioned for example here) and reproducibility, this is a necessary information. Also the (periodic?) boundary conditions of the MCRT are only mentioned later on for the Cb-case.

The information on experimental setup is not organised clearly enough, and some important information is missing. This also hid some decisions and logic of certain setups. To address this issue:

- I now mention that stratus covers half the domain, and why it is effectively periodic (and infinite) in the y-direction
- I mention and motivate the domain size and resolution choices
- I mention when atmospheric profiles are the same between cases, or when they differ. Details of which specific profile it is are less important, and the profile choices are fairly arbitrary. The important part is that moisture profiles affect clear-sky irradiance, and while most presented data is normalized w.r.t. clear-sky, it'll help understand subtle differences in absolute values.
- I changed figure 4b so it plots the 100 m radius version of the cloud gap that is used in the analyses of section 4.2 (Figure 9). Originally, I showed a 500 m radius version, which matched the radius of the plotted cloud disk, but that deviates too much from a small gap as used in Yordanov's work (35 m diameter).
- The simulation tools section (2.1) now mentions the periodic boundary conditions of the ray tracer and that we use a single trace gas profile for all simulation.
- Clear-sky SSI varies between experiments due to differences in atmospheric moisture profiles (notwithstanding the albedo and zenith angle effects), this is clarified in section 2.1, 3.1.1

L. 185:  Fig. 4 suggests the domain is 12.8km x 12.8km for stratus, but 6.4km x 6.4km for cloud gap. Is this "the same configuration" and only an excerpt shown or do the domain sizes differ? Please clarify.

See previous comment.

L. 189: Two times "manually": I suggest deleting the later occurrence

Fixed.

L. 196-201: As later use of this case suggests, the domain size here is not 12.8km x 12.8km (or 6.4km x 6.4km), but larger to isolate the towering cumulus. Please mention the actual

domain size also here. While it is later on (Section 4.5) noted, that the optical thickness is ensured to be large enough, a thorough documentation of your scaling of optical thickness with increasing cloud depth is missing and should be added, for example here.

These are all good points and are now discussed in the case description.

L. 208-209: Repetitive use of "altocumulus", e.g., the second occurrence could be replaced by 'This'.

I have left it unchanged for clarity.

Fig. 6: As cloud depth is later on (e.g. Fig. 13) used to distinguish cases, it would be nice if this would be given for the displayed snapshots for a better association.

Nice suggestions, I have added the cloud depth for the snapshots. Estimating from just the y-axis is indeed not easy.

L. 244: I am unsure whether there should be a second "in" in "for the scattering regime we are [in] in terms of mechanism".

Me neither. I have rephrased the sentence to avoid the awkward double 'in'.

Fig. 9: The domain cross-section used for this plot could be indicated in Fig. 4 to get a feeling of the averaged region in y-direction, especially in the checkerboard case. This would also give a hint on sun direction there. Otherwise "(part of)" (Fig. 9, caption L. 2) is a very vague definition.

Indeed, this was not well explained. I have added a magenta shading to Figure 5d showing the same selection as was used for subsetting and averaging the data in Figure 9cfi, and explained this in the captions.

Fig. 9: While theoretically reconstructable from Norm. SSI (-), AE (W/m$^2$) and AE as % of IE for the reader, it would be helpful for the reader to have clear-sky SSI (dir/dif/tot) values given at least for this figure or more in general in the text.

Clear-sky SSI is a function of atmospheric profile, surface albedo, and solar zenith angle. There will be too many different numbers to give, and normalisation solves this issue. But I agree that there should be some mention of absolute clear-sky SSI to help the reader, so I have added them for the total SSI in the middle row of Figure 9 for each albedo, case, and zenith angle.

Fig. 9 g-i: The extremal values on the shadow borders are a striking feature and should at least briefly assessed in the text. Also, there is no explanation of the grey regions here.

These happen due to IE approaching zero and switching sign from sunlit to shaded areas. This leads to physically meaningless artefacts at the transition and negative values for IE in shaded areas. The gray mask was meant to mask the shaded areas to avoid having to discuss what negative values of "% of IE" mean, but I forgot to explain the masking itself. This is now fixed. The colormap is changed to also make it clearer that the middle and bottom rows are different units.

For clarity, a negative "% of IE" is due to IE being negative in cloud shadows. It is then an irradiance reduction instead of enhancement, and you can still calculate what % of the negative IE is accounted for by increase in irradiance due to albedo.

Fig. 9 i:In the case of sun zenith angle SZA=30° the gray shaded area including the border grid boxes do have an x-extent of about 2x the extent of the valid data area excluding the border grid boxes with extremal values. To my understanding, of the text, the cloud disks are 500m in diameter with a maximum distance of 150m in between (L. 193-195). This would suggest a ratio of > 3:1. I guess this is due to the selected and averaged y-axis range, but this needs more explanation.

It is indeed due to the selected y-axis range. Please refer to the revised figure 4 which better shows how data was selected. My comments on 'maximum' 150 m spacing in between are wrong in hindsight, but it may have been a typo: the spacing is 250 m at most. I have clarified the text.

L. 304-310: A full description of the setup would in my opinion include albedo and SZA in the text, not just in Fig. 10

Fair enough, I have more completely described the setup in text now.

Fig. 10 and Section 4.4.1:

Is the SSI summed/averaged over the entire domain (almost) constant for all cloud altitudes and therefore the power only redistributed or is there a significant difference in surface solar power based on cloud altitude? To me, this would be a nice additional information and support the previous explanation of checkerboard case SSI.

It's nearly constant. Domain averaged SSI changes at most 0.05% (highest vs. lowest cloud, for the 1000 m case). I added this information to the text.

L. 326: Can omit the "in" in "very small [in] for the simulated altocumulus"

Thanks, fixed.

L. 336: "There are two exceptions where forward escape still occurs": In a statistical sense, individual photons/rays can always be scattered only once or twice and therefore there is

always some fraction of forward escape. While the meaning is fully understandable in this context, please consider rephrasing to avoid the impression of exclusivity of the mechanisms. I would suggest replacing "occurs" by 'plays a significant role' or 'contributes significantly'.

Fixed.

Fig. 13 caption:

Neither a nor b do directly show SSI patterns as indicated by the caption. Please make the caption, especially the first sentence, more precise.

This was not well-captioned, indeed. A pattern can (in my view) still be 1D, or a line, so I keep the word 'pattern' for 13a. But it's not an SSI pattern, rather it's an IE pattern, technically.

Fig. 13 a: Are 17.2 and 18.4 ordered on purpose like this in the legend?

It's where time and cloud depth de-couple: 18.4 km is an overshooting top, 17.2 km is where the updraft settles in a later time step. The legend is ordered as function of time. It's why the last scatter points on each line in Figure 13c also decrease in cloud depth (but increase in diffuse enhancement, as the anvil is spreading out). I have added that the cloud top decreases in the last time step for clarity in the caption.

L. 371: There is still a maximum in IE in these cases, it is just spatially shifted and/or decreased. Please rephrase "disappearance" or specify the "peak irradiance enhancement" you mean more in detail.

I now explicitly refer to the 'narrow peak near the cloud edge' as opposed to any local IE maximum.

L. 377: For example 'hinder' or 'obstruct' would describe the process more factual than "takes over the side escape mechanism". At the moment this sentence does not reflect the actual process to me.

Agreed. 'Hinders' fits well here, I have changed the text.

Fig. 14: Cb cases were before referenced by LES time or cloud depth. For better orientation, it would be helpful to get this information here as well.

Added the timestamps to the subplots.

Fig 15 caption, L.3:

"The [S]un"

Fixed.

L. 399-400: The regions meant here are the ones away from 'cloud shadows' and not from "clouds", I suggest? Please consider rephrasing.

I don't quite follow. I am referring in the text to regions on the sunlit side of clouds that are not directly shaded by other clouds.

L. 409: "sunlit cloud base" sounds to me only possible with SZA > 90°. Please clarify.

I changed it to 'cloud base edge'. Cloud base to me is not simply the underside of a cloud, but includes some finite thickness and therefore also a side, but hopefully this new phrasing clarifies my meaning.

L. 427: Also for clouds with optical depth > 6, a (small) fraction of light is scattered only once or twice. Therefore I suggest adding "irradiance is [predominantly] scattered uniformly downward".

Changed, also changed 'uniformly' to 'diffusely' (as per your next comment).

L. 427: To my knowledge, the downward radiance distribution underneath optically thick clouds is not "uniform". Please clarify.

This can be looked up, e.g., in theory and simulation in Sobolev (2017), Grant et al. (1995), in measurements in Nagata et al. (1997) or simply using a 1D-RT calculation with an optically thick cloud.

Changed to 'diffusely'. I was using 'uniform' in the context of the idealised clouds that were uniform by design, but of course that not the case anymore.

L. 439: "onto" seems the wrong word here to me. Perhaps use 'of' instead?

Fixed.

L. 460: Can omit "out".

Done.

L. 469: Following the sentence structure, I believe it should be 'maximally complex' instead of "maximum complexity".

I think so too. I changed the text.

**References**

Sobolev, V. V. (2017). Light scattering in planetary atmospheres: international series of monographs in natural philosophy (Vol. 76). Elsevier.

Grant, R. H., Heisler, G. M., & Gao, W. (1996). Photosynthetically-active radiation: sky radiance distributions under clear and overcast conditions. Agricultural and Forest Meteorology, 82(1-4), 267- 292.

Nagata, T. (1997). Radiance distribution on stable overcast skies. Journal of Light & Visual Environment, 21(1).

---

## Author Response (AR2)

**Referee #1**

I now recommend only one further refinement. Specifically, the paper outline in Line 53 starts with "Next, in Section 2.2, we will...", which suggests that a previous sentence that mentions Section 2.1 is missing. Therefore, a new sentence should be added here to describe what Section 2.1 does.

Thanks for spotting this one, "Section 2.2" should have been "2.1 and 2.2", something went wrong with the section labels after re-organizing the introduction structure.

**Referee #2 (Philipp Gregor)**

The manuscript is improved and I believe that the authors have done a good job in addressing the comments. The described bugs in the raytracing code related to ice treatment are severe, however seemingly of small influence on the results and not apparent when viewed from outside. I appreciate that the authors corrected this.

I have a few additional comments, which I hope the authors will address.

Thank you once again.

Please find our response to your comments below in the blue text.

**General Comments:**

While the description of settings is by far more detailed in the revised manuscript, some details are often only given in Figure captions and not in the textual description of the results. Examples are albedo 0 for the altocumulus fields in 5.4.2 only given in Fig. 11 or in section 5.1 the SZA 45° only given in Fig. 8. I suggest revising this for a complete and sorted description of the experiments in the text.

Albedo of 0 in figure 11 refers to that figure specifically, as we run the altocumulus also with 0.8 to test the effect of albedo (last paragraph of section 5.4.2.). However, we now explicitly state that all simulations are run with an albedo of 0 by default, unless stated otherwise (in which case the effect of albedo is being tested). For clarity I have added the SZA and albedo settings in a handful more occasions where these specifics were only part of the figure caption.

While I do believe the authors based on the presented results, that the Monte Carlo raytracing was run to a point where the noise does not systematically affect the conclusions and results, noise is still very obvious in the figures. While re-running experiments, a standard deviation should have been easy to compute alongside and given in the manuscript. I think it would be helpful to at least mention the noise when introducing

the raytracer. The decreased resolution of the towering cumulus case (Lines 288-290) indicates that runtime in comparison to noise may indeed have been a point of consideration.

The reason we have to reduce the resolution is due GPU memory limitations, not computational time. A 4x increase in resolution would require 4x4x4=64 x more GPU memory for a similar domain size. Multi GPU is not yet supported for this experimental setup, so we simply have had to accept lower resolution.

We think that while running with many more rays will further reduce statistical noise, it does not reduce the true uncertainty of the experiments, and so providing uncertainty based on Monte Carlo noise can be a bit misleading.

We disagree that the noise is 'very obvious', except in Figure 8b. However, we do not discuss noise anywhere in the text. We have added "As for the number of rays, or sample per pixel, we set to this to a high enough to get a clear signal, typically 256 to 1024." to section 3.1

**Specific comments:**

L. 113-114: This is a bit confusing to me. To my understanding the growth of the cumulus congestus causes the shadow to appear while the reappearance in the time series refers to the reappearance of direct irradiance (the Sun). Could you formulate this a bit clearer?

This ambiguity has been fixed.

L. 117: "away from clear-sky region[s]."

Fixed.

L. 291: "naturally influenced [by] turbulence"

Fixed.

L. 257: "is effectively infinite [as] long as"

Fixed.

Section 4.1.1., Fig. 4, or 5.2: Please indicate the averaged y-region for the cloud gap case.

Added text to explain this. The subset is so small that it won't be readable in the figure. It's the center 150 meters of the 200 meter diameter circle.

L. 360: "... can rarely irradiance enhancement ..." probably not correct English

"can rarely [cause]" - Fixed.

L. 360: "the cloud only [??] meters thin"

"is" - I think I was making a summation with the previous "is" but that's not correct English... Thanks.

L. 435-436: "... direct beam location would focus on one spot when." Sentence seems to miss second half.

"when" can be removed, sorry, and thanks for spotting this one.

L. 437-438: "magnitude of such effects [is] hard to estimate"

Indeed, fixed.

L. 442: "solar zenith angle of 0[°] unrealistic."

Fixed.

Fig. 12c: y-Axis unit: Is "W m^-2" correct here or should it be "%" as in the following figures?

It is correct, but I understand your question. 4 W/m^2 is very little, but it's averaged over a large area where nothing happens.

L. 477: "We identify [...] three distinct"

Fixed.

Fig. A4, A5,A6: The naming of the number concentration experiments is inconsistent between Figures and captions (nc200, ncx2,nc_x2, ...). Also, the titles of the a) and b) subfigures is irrelevant. Please Fix.

I made the captions consistent with the figures. I don't think it hurts to leave the subfigure titles up.